# Phytochemical Profiling, Biological Activities, and In Silico Molecular Docking Studies of *Causonis trifolia* (L.) Mabb. & J.Wen Shoot

**DOI:** 10.3390/plants12071495

**Published:** 2023-03-29

**Authors:** Samik Hazra, Anindya Sundar Ray, Swetarka Das, Arunava Das Gupta, Chowdhury Habibur Rahaman

**Affiliations:** 1Ethnopharmacology Laboratory, Department of Botany, Visva-Bharati University, Santiniketan 731235, West Bengal, India; 2Department of Animal Science, Kazi Nazrul University, Asansol 713340, West Bengal, India; 3Division of Microbiology, CSIR-Central Drug Research Institute, Lucknow 226031, Uttar Pradesh, India; 4Division of Microbiology and Immunology, College of Veterinary Medicine, Cornell University, Ithaca, NY 14853, USA

**Keywords:** *Causonis trifolia*, response surface methodology, phytochemical profiling, acute toxicity, molecular docking

## Abstract

*Causonis trifolia* (L.) Mabb. & J.Wen, commonly known as “fox grape”, is an ethnomedicinally important twining herb of the Vitaceae family, and it is used by ethnic communities for its wide range of therapeutic properties. Our research aims to investigate the chemical composition; antioxidant, anti-inflammatory, and antidiabetic activities; and mechanisms of interaction between the identified selective chemical compounds and the target proteins associated with antioxidant, anti-inflammatory, and antidiabetic effects of the optimised phenolic extract of *Causonis trifolia* (L.) Mabb. & J.Wen, shoot (PECTS) to endorse the plant as a potential drug candidate for a future bioprospecting programme. Here, we employed the response surface methodology (RSM) with a Box–Behnken design to enrich the methanolic extract of *C. trifolia* shoot with phenolic ingredients by optimising three key parameters: solvent concentration (% *v/v*, methanol:water), extraction temperature (°C), and extraction duration (hours). From the quantitative phytochemical estimation, it was evident that the PECTS contained good amounts of phenolics, flavonoids, tannins, and alkaloids. During the HPLC analysis, we identified a total of eight phenolic and flavonoid compounds (gallic acid, catechin hydrate, chlorogenic acid, caffeic acid, p-coumaric acid, sinapic acid, coumarin, and kaempferol) and quantified their respective contents from the PECTS. The GC-MS analysis of the PECTS highlighted the presence of 19 phytochemicals. In addition, the bioactivity study of the PECTS showed remarkable potentiality as antioxidant, anti-inflammatory, and antidiabetic agents. In silico molecular docking and computational molecular modelling were employed to investigate the anti-inflammatory, antioxidant, and antidiabetic properties of the putative bioactive compounds derived from the PECTS using the GC-MS technique to understand the drug–receptor interactions, including their binding pattern. Out of the 19 phytocompounds identified by the GC-MS analysis, one compound, ergosta-5,22-dien-3-ol, acetate, (3β,22E), exhibited the best binding conformations with the target proteins involved in anti-inflammatory (e.g., Tnf-α and Cox-2), antioxidant (SOD), and antidiabetic (e.g., α-amylase and aldo reductase) activities. The nontoxic nature of this optimised extract was also evident during the in vitro cell toxicity assay against the Vero cell line and the in vivo acute toxicity study on BALB/c mice. We believe the results of the present study will pave the way for the invention of novel drugs efficacious for several ailments using the *C. trifolia* plant.

## 1. Introduction

For thousands of years, nature has been a source of medicinal agents, and a remarkable number of modern medicaments have been obtained from natural sources, mainly from plants, and many of them are based on their usage in folk medicine systems. In the present scenario, the demand for herbal products is growing exponentially throughout the world, and scientists are paying attention to the screening of different pharmacological activities of plant extracts and isolating biologically active substances for their potential medicinal value.

*Causonis trifolia* (L.) Mabb. & J.Wen, commonly known as “fox grape” (synonym: *Cayratia trifolia* (L.) Domin), is an ethnomedicinally important twining herb belonging to the Vitaceae family. This perennial climber is native to Australia and several countries in Asia, including India. In India, this herb grows in the wild in Jammu, Rajasthan, Assam, Tripura, and West Bengal, extending into peninsular India by up to 600 m [1]. The broad-spectrum medicinal value of this herb is well documented in several important studies and practiced in various traditional communities for curing a wide range of health hazards. The whole plant is used as a diuretic and in tumours, neuralgia, and splenopathy [2,3]. Leaf decoctions, or the juice of the fresh leaves, are used to cure scurvy [4]. The roots are used as an astringent [5] and as an antidote to snake bites [6]. To control blood sugar levels, diabetic patients are administered with an aqueous extract of the tuber along with the infusion of seeds, whereas the powdered root tuber is administered orally with milk for the early healing of fractured bones [6,7]. Very few scientific explorations have been conducted previously to investigate the phytochemical and pharmacological potential of the studied plant parts. In 2018, Yusuf and coworkers investigated the in vitro antidiabetic potential of an ethanolic stem extract against a mice model. The study showed that the methanolic stem extract had a strong hypoglycaemic effect at a dose of 650 mg/kg of body weight [8]. The in vitro antioxidant and antimicrobial potentials of an n-hexane whole plant extract of this herb were highlighted in recent experiments by Meganathan et al. (2021) [9]. Moreover, several groups of scientists have performed a preliminary phytochemical screening and TLC and HPTLC studies of various crude extracts of the different parts of this plant [4,8,10,11]. However, there are some gaps in our knowledge of how the chemical compounds present in this plant are closely connected to the therapeutic properties ascribed to it and how they affect each other. In our research, we used an integrated in vitro–in silico strategy to predict the lead molecules that are mainly responsible for the *Causonis trifolia* plant’s potential as an antioxidant, anti-inflammatory, and antidiabetic agent. Meanwhile, there are no previous reports on the optimisation of the extraction process for obtaining the most efficient condition for the extraction of the maximum yield of the bioactive phenols of this plant. Here, we made an attempt to optimise the extraction process using a suitable statistical tool.

An accurate extraction process is regarded as the most critical step behind the isolation of higher amounts of bioactive phytocompounds from herbal drugs. Like other phytochemical groups, the extraction of phenols from the powdered biomass of plants is highly influenced by the extraction environment, such as the temperature and solvent composition. Traditional phenol extraction methods employing one factor at a time have several drawbacks, such as a prolonged extraction time, unsatisfactory extraction efficiency, and breakdown of the thermolabile compounds. Furthermore, the interactions between the different factors in a single experimental setup are usually overlooked, resulting in a very low probability of attaining a true optimum. To address these issues, the response surface methodology (RSM) was developed to optimise the levels or values of the multiple extraction factors to achieve the best performing system for the maximum extraction of bioactive phenols from this plant’s parts [12]. In the present study, we report an optimised extraction method for the extraction of total phenols from the shoots of *Causonis trifolia*.

The broad-spectrum therapeutic properties of this ethnomedicinal herb can be attributed to the large reservoir of bioactive secondary metabolites that occurs in this plant species. The HPLC and GC-MS techniques have firmly established themselves as key techniques for the phytochemical profiling of herbal drugs [13]. Phytochemical screening not only reveals the contents and types of constituents of plant extracts, but it also aids in the search for bioactive compounds that may be utilised in the development of effective drugs. Therefore, our present experiment focused on the quantification of major phytocompounds, such as phenol, flavonoid, tannin, and alkaloid, as well as the identification of bioactive compounds from an optimised phenolic extract of *Causonis trifolia* shoot (PECTS), employing HPLC and GC-MS analyses. Moreover, the in vitro biological activity studies provide basic and fundamental information on the pharmacological properties of drug elements, and these preliminary reports play a very significant role in the validation of the ethnomedicinal claims while also curing various diseases using this studied herb. Keeping this in mind, we also evaluated the therapeutic potentiality of PECTS, employing in vitro antioxidant, anti-inflammatory, and antidiabetic investigations.

Natural products with multiple pharmacological actions have been demonstrated as potential therapeutic candidates for treating a wide range of health complaints. Unfortunately, there are significant gaps in our understanding of which chemicals interact with which targets, and evaluating all probable interactions experimentally is impractical and also needs large amounts of time, expensive infrastructural facilities, and mammoth funding. Recent advances and developments in computational (in silico) approaches provide powerful tools for drug discovery, which can be applied to explore the poly-pharmacological profiles of phytochemicals found in medicinal plants [14]. Protein–ligand interactions are analogous to the lock-and-key paradigm in which the lock represents the protein and the key is correlated with the ligand (i.e., the bioactive compound). This entails docking a library of phytocompounds/ligands into a biological target and calculating the probability that a ligand will bind to the protein target using a scoring algorithm, which aids in the discovery of the most promising lead compounds for a biological target. In this regard, in silico molecular docking and computational molecular modelling were employed to investigate the anti-inflammatory, antioxidant, and antidiabetic properties of putative bioactive compounds derived from PECTS using GC-MS analysis.

Herbal products are often assumed to be safe or low in toxicity due to the fact of their long history of usage by humans [15]. Nevertheless, the latest studies indicate that many of these medicinal plants used for curing a wide range of diseases are potentially toxic, carcinogenic, or mutagenic when they are administered at high doses for prolonged periods [16,17,18]. As safety is still a major concern with the use of medicinal plants, screening for possible toxicity and the determination of permissible doses of herbal products are worthwhile endeavours. Even though *Causonis trifolia* is used in alternative medicine systems and enjoys large popularity globally due to the fact of its wide therapeutic applications, unfortunately, adequate information on its toxicity profile is still not available. In our present study, as part of a safety evaluation of PECTS, an in vitro cell toxicity assay against the Vero cell line and an in vivo acute toxicity study on BALB/c mice were performed.

In summary, our research objective was to statistically optimise the extraction procedure to obtain the maximum extraction of bioactive phenols, investigate the chemical composition, and explore the in vitro antioxidant, anti-inflammatory, and antidiabetic potential of the optimised plant extract and the mechanisms of the interactions between the identified chemical compounds via GC-MS analysis and the target proteins associated with the antioxidant, anti-inflammatory, and antidiabetic effects. Our investigation also explored both the in vitro and in vivo toxicity profiles of the optimised phenolic shoot extract to endorse the plant for a rational bioprospecting programme as a potential drug candidate.

## 2. Results

### 2.1. Statistical Optimisation of the Extraction Process for the Enhanced Extraction of Phenols

In our present study, linear, interactive (2 FI), quadratic, and cubic models were fitted to the experimental data to create regression models, and the results are provided in Appendix A. The quadratic model was the most adequate model for representing our experimental data set. An ANOVA was used to examine the RSM’s statistical optimisation of the extraction parameters (analysis of variance). An ANOVA was also utilised for the regression analysis, results verification, and case statistics prediction (see Appendix A). The Model F-value of 116.87 implies that the applied model is quite significant. A small F-value for the model is not desired, since it indicates that the variance was caused by random unexplained disturbances, referred to as noise. The probability of an F-value of 116.87 occurring due to the fact of noise is only 0.01%. *p*-Values less than 0.0500 indicate that the model terms are significant. In this case A, BC, A^2^, B^2^, and C^2^ are significant model terms. If there are many insignificant terms (not counting those required to support the hierarchy), the model reduction can improve the model’s quality. The lack-of-fit test was used as a supporting test for the adequacy of the fitted model. A lack-of fit F-value of 1.95 implies that the lack of fit is not significant. A nonsignificant lack of fit is good, as we want the model to fit. The model showed a standard deviation, R^2^, and mean of 2.76, 0.9934, and 119.54, respectively. The predicted R^2^ value of 0.9330 is in reasonable agreement with the adjusted R^2^ of 0.9849, i.e., the difference was less than 0.2, which again indicates the validity of the model. “Adequate precision” measures the signal-to-noise ratio, and if it is greater than 4, then the model has a signal strong enough to be used for optimisation. The ratio of 25.237 indicated an adequate signal. Here, a lesser value for the coefficient of variation (CV%; 2.31) designates a better reliability of the model, whereas a correlation coefficient (R^2^) of 0.9934 indicates a high degree of correlation between the experimental parameters and the response [19]. The three-dimensional response surface plots depict the interaction among the three parameters studied and demonstrate the combined influence of the parameters on the extraction of total phenolics from shoots of *C. trifolia*. The effect of their variables and interaction on the responses can be seen in Figure 1A–C. Therefore, an accurate optimisation of the level of each parameter is essential to obtain the maximum amount of phenolics. A normal probability plot is shown in Figure 1E, displaying scatters present closely along the line, indicating that the residuals followed a normal distribution, satisfying the assumptions of the ANOVA, which designated the accuracy of the RSM in optimising the three variables [20]. Similarly, as shown in Figure 1F, it is suggested that the actual and predicted values of the TPC were in close agreement, which again validates the model. Finally, the predictive extraction model suggests that a methanol concentration of 67.6754%, extraction temperature of 49.7753 °C, and extraction time of 12.5837 h are the most suitable and effective combinatory parameters for obtaining the highest yield of bioactive polyphenols (152.774 mg GAE/g dry tissue) with a 94.5% desirability, which is well accepted (Figure 2).

### 2.2. Quantitative Estimation of Certain Groups of Phytochemicals of the PECTS

The optimised values of the independent variables were used for the extraction, and the total phenolic content (i.e., response) was compared with the value generated by the RSM. The total phenolic content of the optimised phenolic extract of *Causonis trifolia* shoot (PECTS) was estimated to be 150.76 ± 0.02 mg GAE/g dry tissue, which is very close to the predicted value of the RSM model (152.774 mg GAE/g dry tissue). The contents of total flavonoid, total tannin, and total alkaloid of the PECTS were 20.7 ± 0.02 mg CE/g dry tissue, 23.21 ± 0.47 mg of TAE/g, and 13.37 ± 0.12 mg of PE/g, respectively.

### 2.3. High-Performance Liquid Chromatography (HPLC) Analysis of the PECTS

For the HPLC analysis, a total of 10 standards of phenol and flavonoid compounds (gallic acid, kaempferol, catechin hydrate, chlorogenic acid, caffeic acid, syringic acid, p-coumaric acid, sinapic acid, quercetin, and coumarin) were used. Qualitative and quantitative evaluations of these therapeutically important phytochemicals from the optimised shoot extract of *Causonis trifolia* were undertaken in the present experiment. this study identified a total of eight compounds of phenols and flavonoids (gallic acid, catechin hydrate, chlorogenic acid, caffeic acid, p-coumaric acid, sinapic acid, coumarin, and kaempferol), and the compounds were quantified through an HPLC analysis of the PECTS (Figure 3). This study also highlights that considerably higher amounts of sinapic acid (24.172 mg/g dry tissue), coumarin (13.211 mg/g dry tissue), and gallic acid (7.766 mg/g dry tissue) were estimated in the PECTS (Figure 4).

### 2.4. Gas Chromatography-Mass Spectrometry (GC–MS) Analysis of PECTS

The results pertaining to the GC-MS analysis led to the identification of 19 compounds from the GC fractions of the optimised phenolic extract of *Causonis trifolia* shoot (PECTS) (Figure 5). A detailed list of the identified compounds along with their chemical characters and peak areas are presented in Table 1. The 19 phytocompounds identified in the PECTS are 5,7-dodecadiyne-1,12-diol (1.123); ergosta-5,22-dien-3-ol, acetate, (3β,22E) (1.167); 1,2,4-metheno-1H-indene, octahydro-1,7a-dimethyl-5-(1-methylethyl)-, (1S-(1α,2α,3aβ.,4 α.,5α,7aβ.,8S*)) (13.403); alpha-copaene (13.784); 1,5-cyclodecadiene, 1,5-dimethyl-8-(1-methylethenyl)-, (S-(Z,E)) (14.305); 3H-3a,7-methanozulene, 2,4,5,6,7,8-hexahydro-1,4,9,9-tetramethyl-, (3aR-(3aα,4β,7α)) (14.385); caryophyllene (14.981); 1H-cyclopropa[a]naphthalene; 1a,2,3,5,6,7,7a,7b-octahydro-1,1,7,7a-tetramethyl-, (1aR-(1aα,7α,7aα,7bα)) (15.720); humulene (15.955); 1,4-methano-1H-indene, octahydro-4-methyl-8-methylene-7-(1-methylethyl)-, (1S-(1α,3aβ.,4α,7α,7aβ)) (16.397); 1H-3a,7-methanoazulene, octahydro-1,9,9-trimethyl-4-methylene-, (1.α.,3aα.,7 α.,8aβ) (16.534); germacrene D (16.781); naphthalene, 1,2,4a,5,6,8a-hexahydro-4,7-dimethyl-1-(1-methylethyl) (17.433); 1-isopropyl-4,7-dimethyl-1,2,3,5,6,8a-hexahydronaphthalene (18.089); 11,11-dimethyl-spiro [2,9] dodeca-3,7-dien (19.549); 2H-pyran, tetrahydro-4-methyl-2-(2-methyl-1-propenyl) (22.058); 3-methylmannoside (23.140); 3-O-methyl-d-glucose (23.573); and γ-elemene (38.188) (Figure 6). The highest and lowest peak areas were found in the case of germacrene D (16.755%) and 3-O-methyl-d-glucose (0.286%), respectively. The reported biological activity of all of the detected compounds are summarised in Table 2.

### 2.5. In Vitro Antioxidant Study of the PECTS

#### 2.5.1. DPPH Radical Scavenging Activity

In this study, different concentrations of the optimised PECTS were subjected to a 2,2-diphenyl-1-picryl-hydrazyl-hydrate (DPPH) free radical scavenging assay. Figure 7A shows the DPPH free radical scavenging activity of the PECTS along with the standard of ascorbic acid. At a concentration of 100 μg/mL, the scavenging activity of the PECTS was 88.27 ± 0.57%, whereas, at the same concentration, 89.68 ± 3.14% DPPH radical scavenging activity was shown by the ascorbic acid. The IC_50_ values of the PECTS and ascorbic acid were calculated as 59.96 μg/mL and 28.99 μg/mL, respectively.

#### 2.5.2. Hydrogen Peroxide Scavenging Assay

The hydroxyl radical scavenging activity of the PECTS sample was dose dependent (Figure 7B). At a concentration of 100 µg/mL, the hydroxyl radical scavenging activity of the PECTS and the standard ascorbic acid was 69.87 ± 2.29%, and 88.93 ± 2.47%, respectively. The IC_50_ value of the PECTS was estimated to be 68.13 μg/mL. An amount of 35.68 ± 1.16 μg/mL ascorbic acid was needed to obtain a 50% inhibition of the hydroxyl scavenging activity.

#### 2.5.3. ABTS Radical Scavenging Activity

The relative antioxidant ability of the PECTS to scavenge the radical ABTS^+^ was compared with the standard of ascorbic acid. The ABTS radical cation was produced in the stable form using potassium persulphate. After obtaining a stable absorbance, an extract was added to the reaction medium, and its antioxidant power was measured by studying the decolourisation of the medium. The IC_50_ value of the PECTS was estimated to be 47.94 µg/mL. At a concentration of 100 µg/mL, the PECTS exhibited 59.31 ± 0.21% ABTS^+^ radical scavenging activity. At 100 µg/mL, the ascorbic acid showed the highest scavenging activity of 85.22 ± 1.91%, with an IC_50_ value of 33.51 µg/mL (Figure 7C).

#### 2.5.4. Phosphomolybdenum Assay

The total antioxidant capacity (TAC) was based on the reduction of the valency of molybdenum from 6 to 5 by the extract and subsequent formation of the green phosphatemolybdenum complex. It was employed to evaluate the total antioxidant capacity of antioxidants soluble in water and fat. The total antioxidant capacity of the optimised phenolic extract of *Causonis trifolia* shoot was found to be 40.20 ± 1.57 mg AAE/g.

### 2.6. In Vitro Anti-Inflammatory Study of the PECTS

#### 2.6.1. Inhibition of the Albumin Denaturation Assay

The denaturation of proteins is a well-known cause of cell inflammatory responses [30]. BSA was used as a reagent for the assay. BSA accounts for approximately 60% of the total protein in animal serum. BSA undergoes denaturation upon heating and starts expressing antigens associated with Type III hypersensitive reaction, which can lead to illness, such as glomerulonephritis and rheumatoid arthritis [31]. Thus, the inhibition of the BSA denaturation assay was used to evaluate the anti-inflammatory potential of the optimised phenolic extract of *Causonis trifolia* shoot (PECTS).

The extract represented a dose-dependent inhibition of the BSA denaturation (Figure 8A). The IC_50_ values of the PECTS and standard drug (diclofenac sodium) were determined to be 306.78 μg/mL and 289.74 μg/mL, respectively.

#### 2.6.2. Antiprotease Activity

The role of proteases in the pathophysiology of arthritis has been well documented. Neutrophils are reported to be a rich source of serine proteases, which are localised in lysosomal granules. Upon inflammatory stimulation, these serine proteases are released into extracellular spaces by exocytosis, and their uncontrolled release can lead to various chronic inflammatory disease conditions and cause damage to tissues. Protease inhibitors can provide substantial protection against this effect [32]. In our assay, trypsin was utilised as a serine protease enzyme and casein was used as a substrate.

The optimised extract of *Causonis trifolia* exhibited a concentration-dependent antiprotease action comparable to the antiprotease activity of the conventional drug at higher doses (Figure 8B). The IC_50_ value of the PECTS was 290.97 μg/mL, and it was nearer to the value of 270.19 μg/mL for diclofenac sodium, the standard drug.

#### 2.6.3. Membrane Stabilisation Assay

As the membrane of human red blood cells (HRBCs) is similar to that of a lysosomal membrane in terms of integrity, it was employed to assess the membrane stabilising capacity of a drug or extract. If the extract can stabilise the HRBC membrane, it may be able to inhibit the release of lysosomal enzymes from the lysosome by stabilising its membrane system. Because heat induces membrane lysis and haemoglobin oxidation, the anti-inflammatory activity of the plant extract was studied using the membrane stability of HRBC against heat [33].

##### Heat-Induced Haemolysis Assay

The optimised phenolic extract of the PECTS exhibited a dose-dependent inhibition. The PECTS exhibited an IC_50_ value of 327.041 μg/mL, while the standard drug (diclofenac sodium) depicted more potency with an IC_50_ value of 253.48 μg/mL (Figure 8C).

### 2.7. In Vitro Antidiabetic Study of the PECTS

#### 2.7.1. α-Amylase Inhibitory Assay

α-Amylase is an intestinal enzyme that degrades starch molecules into smaller units of two or three glucose monomers by acting upon the α-1,4-glycosidic bonds present in the starch polysaccharide. The suppression of digestive enzymes such as α-amylase causes a delay in the digestion of starch and oligosaccharides, which reduces glucose absorption and, as a result, lowers the blood glucose level [34]. The optimised phenolic extracts of *Causonis trifolia* (PECTS) were subjected to an α-amylase inhibitory assay along with acarbose as a standard (Figure 9A). The PECTS exhibited a percentage inhibition of 68.42 ± 0.13% at a 200 µg/mL concentration with an IC_50_ value of 117.38 μg /mL. The acarbose, a standard antidiabetic drug, showed an α-amylase inhibitory activity of 77.2 ± 0.27% at 200 µg/mL with an IC_50_ value of 84.48 µg/mL.

#### 2.7.2. Yeast Cell Glucose Uptake Assay

The glucose transport mechanism across the yeast cell membrane has received attention as a tool for the in vitro screening of different medicinal plant extracts for their hypoglycaemic effect. The amount of glucose that remains in the medium after a specific amount of time serves as a marker of the yeast cell glucose uptake. It has been reported that in yeast cells (*Saccharomyces cerevisiae*), glucose is transported by a facilitated diffusion process. Facilitated carriers are specific carriers that transport solutes from the higher concentration region to the lower concentration region. This means that effective transport will only be attained if there is the removal of intracellular glucose. In the present study, the optimised phenolic extract of *Causonis trifolia* (PECTS) were subjected to an in vitro yeast cell glucose uptake assay. The percentage of the increase in the glucose uptake in the yeast cells by the action of the PECTS was compared with the standard drug, acarbose. The increased concentration of the PECTS corresponded with the increased percentage of glucose uptake in the yeast cells. This result indicates that high concentrations of PECTS exhibit high glucose uptake (Figure 9B). The PECTS exhibited the highest percentage of glucose uptake: 55.07 ± 0.25%, at a 200 µg/mL concentration. The IC_50_ value of the PECTS was estimated to be 156.42 µg/mL. With an IC_50_ value of 52.95 µg/mL, the acarbose showed a higher potency.

### 2.8. In Silico Molecular Docking Study

A total of 19 bioactive chemical compounds were identified through a GC–MS analysis of the optimised phenolic extract of the investigated plant. These 19 chemical compounds were then analysed for their activities against the target proteins associated with inflammatory, oxidative, and diabetic conditions. To determine the binding affinities of the 19 phytocompounds to five target proteins, docking studies were performed using the iGEMDOCK 2.1 software. Among the 19 phytocompounds, 1 compound, Ergosta-5,22-dien-3-ol, acetate, (3β,22E), exhibited the best binding conformations with the lowest binding energy values with all five target proteins, such as inflammatory (e.g., Tnf-α (−93.0843 kcal/mol), Cox-2 (−92.0437 kcal/mol)), oxidative (e.g., SOD −87.7343 kcal/mol), and diabetic (e.g., α-amylase (−108.844 kcal/mol) and aldo reductase (−115.66 kcal/mol)) proteins (see Appendix A). Our findings are in agreement with the earlier research findings, and they reveal that a lower binding energy score results in a greater protein–ligand binding stability [35,36].

The Autodock4.2 programme was used to determine the relative strengths of the binding interactions of the protein–ligand complex and to analyse the conformation and orientation (referred to together as the “pose”) of the ligands into the binding site of a protein target. The top scoring phytocompound, Ergosta-5,22-dien-3-ol, acetate, (3β,22E), against each target protein was identified and subjected to further docking analysis using the Autodock 4.2 programme. The docking study revealed the potential binding affinity of the most potent phytocompound into the binding sites of the target proteins with minimum binding energies (ranging from −7.16 to −11.73 kcal/mol), ligand efficiency (ranging from −0.22 to −0.37 kcal/mol), and inhibition constant (5.62 to 964.03 nM) (Figure 10; Table 3). This compound also formed hydrogen bond interactions and the best possible binding pose with the residues of the targeted anti-inflammatory, antioxidant, and antidiabetic proteins, as shown by their corresponding 3D interaction models in Figure 11, Figure 12, Figure 13, Figure 14 and Figure 15, respectively.

### 2.9. Toxicity Analysis of the PECTS

#### 2.9.1. In Vitro Cytotoxicity Assay

The in vitro cytotoxicity study of the optimised phenolic extract of *Causonis trifolia* (PECTS) in the Vero cell line (ATCC-CCL-81) did not show any significant cell death, even at the highest dose of 250 mg/L, and the cell morphology was also normal (Figure 16).

#### 2.9.2. In Vivo Acute Toxicity Study

In the in vivo acute toxicity assay, the optimised phenolic extract of *Causonis trifolia* (PECTS) did not show any mortality or sign of toxicity in both males and females at a dose of 5000 mg/kg body weight over the observation period. Moreover, the treated mice did not show any characteristic change in body weight during the experimental period. As no deaths were observed in this experiment at the highest dose of administration, the LD_50_ of the PECTS could not be estimated and was considered to be greater than 5000 mg/kg body weight, an experimental upper limit, in the acute toxicity study (Figure 17A,B). Thus, the high LD_50_ value of this plant is a strong indicator of the fact that the extract could be considered safe.

## 3. Discussion

The use of medicinal plants for curing diseases is as old as humankind itself. Contemporary science has recognised the active role of plants in the treatment of various diseases, and it has incorporated a variety of plant-derived drugs into modern pharmacotherapy that have been used throughout the millennia. The vast diversity of flora on the Indian subcontinent allows for many such plants to be discovered, and *Causonis trifolia* (L.) Mabb. & J.Wen (Family: Vitaceae) is one such underexplored ethnomedicinal plants that has been used by various ethnic groups across the globe including in India.

The extraction procedure can be improved by optimising the different experimental factors that affect the extraction to achieve the highest possible output with bioactive phytoconstituents. Many works have been carried out to standardise an optimum method for the extraction of the highest amounts of bioactive phytoconstituents from a particular herbal drug. Some important examples can be highlighted here: optimisation of the extraction process of bioactive phytocompounds from the leaf of *Berberis vulgaris* [37], withanolides extraction process optimisation from the root of *Withania somnifera*, and optimisation for essential oils yield from the leaf of *Citrus latifolia* [38]. In our present study, major efforts were carried out to optimise the extraction process of *C. trifolia* shoots. As there is no previous study regarding the optimisation of the extraction conditions for the most efficient extraction of phenolic compounds from the shoot part of *C. trifolia*, this study can serve as a reference for obtaining the maximum yield of phenolic compounds from the studied part of the plant. We adopted an RSM-based optimisation using a Box–Behnken design to achieve this goal. The experimental outcomes suggest that the optimum phenolic content can be derived using the optimised extraction module with a 94.5% desirability. Our study has shown that the extraction of greater amount of active phenols is highly dependent on three parameters, namely, the solvent concentration (%), extraction temperature (°C), and duration of contact (hrs). Three-dimensional response surface plots depict the interaction among the three parameters studied and identify the combined influence of these three parameters on the extraction of total phenols from the dried shoot of the investigated plant (Figure 1 and Figure 2). Moreover, the most effective extraction conditions for the highest yield of phenolics were derived from the RSM analysis: 67.6% methanol concentration, 49.7 °C extraction temperature, and 12.5 hrs extraction time.

Phenolic compounds contained in the medicinal plants and various food stuffs represent the most widely distributed (more than 8000 chemical structures known in the phenolic group) plant secondary metabolites exhibiting a plethora of beneficial effects for human health [39,40]. Epidemiological studies have shown that many phenolic derivatives possess strong antioxidant and radical scavenging properties and are found to be instrumental in reducing the risk of acute and chronic diseases, cardiovascular disorders, and certain types of cancer [40]. In our present experiment, significant amount of total phenols (150.76 ± 0.02 mg GAE/g dry tissue) was found in the optimised phenolic extract of *Causonis trifolia* shoot (PECTS). The phenolic content of our studied plant was higher than the contents of phenolics of many well recognised medicinal plants, such as myrobalan (134.47 mg GAE/g dry tissue), ginger rhizome (60.34 mg GAE/g dry tissue), and emblica fruit (81.5–126.0 mg GAE/g dry tissue) [41,42,43]. Among other phytochemical groups, flavonoids, alkaloids, and tannins have been recognised as therapeutically important groups with various key biological functions, such as free radical scavenging, anticarcinogenic, and anti-inflammatory effects [44,45]. From our study, it is evident that the analysed plant part contains quite a good amount of flavonoids (20.7 ± 0.02 mg CE/ g dry tissue), tannins (23.21 ± 0.47 mg GAE/g dry tissue), and alkaloids (13.37 ± 0.12 mg of PE/100 g dry tissue) as well. The presence of these therapeutically active phytochemicals in a very impressive amount also highlights the prospect of this plant as a good source of potent natural products with multiple therapeutic claims.

Moreover, the HPLC analysis of the optimised phenolic extract of the studied plant (PECTS) identified and quantified a total of eight phenolic and flavonoid compounds, namely, gallic acid, catechin hydrate, chlorogenic acid, caffeic acid, p-coumaric acid, sinapic acid, coumarin, and kaempferol. Hydroxycinnamic acid, synthesised by most of plant species, is a very significant member of phenolic acids with a bioactive carboxylic acid group, and it is present in food items, such as coffee, wine, tea, and cabbage, and in many popular herbal medicines, such as basil and turmeric [46,47]. This class of plant phenolics mainly includes caffeic acid, ferulic acid, sinapic acid, and coumarin. The perusal of the literature suggests that these compounds are capable of donating their hydrogen atom for the neutralisation of reactive free radicals. These phenolic acids and their derivatives are claimed to have several health benefits, including anti-inflammatory, cardioprotective, antiviral, and anticarcinogenic properties. A wide range of in vitro and in vivo studies have demonstrated the therapeutic potencies of many compounds in the phenolic acid group [47]. From the HPLC study, it was noticed that the PECTS is a very potent source of two important phenolic acids, namely, sinapic acid (24.172 mg/g dry tissue) and coumarin (13.211 mg/g dry tissue). Sinapic acid is known as a potent antioxidant, and the antibacterial, anti-inflammatory, anticancer, and anti-anxiety properties of this compound are well documented [48]. On the other hand, coumarin and its derivatives have also shown very strong pharmacological effects, including antitumor, anti-inflammation, antiviral, and antibacterial activities [49]. Our HPLC analysis of the PECTS clearly validates certain ethnomedicinal claims of this studied plant by elucidating the considerable amounts of the eight therapeutically potent phenolic molecules, and it also brings this underexplored medicinal plant into the limelight of scientific communities for its thorough chemopharmacological exploration.

Furthermore, the GC–MS analysis revealed the existence of 19 phytochemicals in the optimised phenolic extract of *Causonis trifolia* shoot (PECTS), which might contribute to the medicinal characteristics of this plant species. Out of the 19 phytochemicals identified, 12 compounds fall under the chemical group of sesquiterpenes. Germacrene D; 1,2,4-Metheno-1H-indene; octahydro-1,7a-dimethyl-5-(1-methylethyl); [1S-(1α,2α,3aβ.,4 α.,5α,7aβ.,8S*)]; α-copaene; caryophyllene; and humulene are some of the sesquiterpenes that were detected in the PECTS by the GC–MS study. Germacrene-D, a sesquiterpenoid compound extracted from the leaves of *Chloroxylon swietenia* DC., has been reported for its mosquitocidal, as well as repellent, activity against aphids and ticks. Germacrene-D also possesses antibacterial properties [25,26,27]. 1,2,4-Metheno-1H-indene, octahydro-1,7a-dimethyl-5-(1-methylethyl), popularly known as Cyclosativene, reported from an ethyl acetate extract of *Psidium guajava* leaves, exhibits strong antiproliferative and antioxidant potencies [21]. The presence of the phytocompound α-copaene are considered a chemotaxonomic marker of the genus *Annona*, since their occurrence is very common in species of this genus. α-Copaene, identified in essential oils of *Annona salzmannii* and *A. pickelii*, has been reported to have antiproliferative and antioxidant properties [21,50]. Caryophyllene is a bicyclic sesquiterpene, previously reported in the essential oil of *Aquilaria crassna*, and displayed anticancer, antioxidant, and antimicrobial properties [24]. Humulene is the primary terpene found in hops (*Humulus lupulus*), but it is also present in cannabis, sage, and ginseng. Humulene, commonly known as α-caryophyllene, has shown potent anti-inflammatory activity in the rat models. Humulene is an effective analgesic agent and is known to have antineoplastic effect by inducing apoptosis [28]. 2H-Pyran, tetrahydro-4-methyl-2-(2-methyl-1-propenyl), also known as rose oxide, is the only monoterpenoid compound identified in the PECTS through our GC-MS study. Nonato et al., in 2012, experimentally proved the anti-inflammatory activity of this monoterpenoid compound [22]. Many studies have apprised the remarkable pharmacological effects of plant sterols, acting as chemopreventive, anti-inflammatory, antioxidant, antidiabetic, and antiatherosclerotic agents [51]. Ergosta-5,22-dien-3-ol acetate (3,22E), a sterol compound, has previously been reported in soft coral, (*Subergorgia reticulata*), but its biological activity has yet to be documented [52].

Based on the findings presented above, it can be concluded that the *Causonis trifolia* shoot extract contains a good number of therapeutically important phytochemicals including phenolic aids that have exhibited significant levels of various pharmacological actions, such as antioxidant, antimicrobial, anti-inflammatory, antiproliferative, and anticancer activities. Identifying various phytochemical compounds provides insight into the medicinal values of the plant *Causonis trifolia*, which can be further evaluated through bio-prospecting studies to optimise how this plant can be utilised to explore its therapeutic potential. In addition, we have also provided experimental evidence obtained in the present study in support of the antioxidant, anti-inflammatory, and antidiabetic activities of the optimised phenolic extract of *Causonis trifolia* shoot (PECTS).

The quest for the identification of active principles is a lengthy, tedious process, not to mention its overburdening financial demands. According to a study conducted in 2014, the cost of developing a new drug was estimated to be USD 2.5 billion [53]. The high failure rate of drug candidates is a primary causes of this dramatic increase in cost [54]. Understanding the drug–receptor interaction is crucial for managing the difficulties scientists face regarding the cost and establishment of a new lead molecule. Fortunately, computational tools have unquestionably played a pivotal role in optimising, assuring, and accelerating the drug development process. Molecular docking is now considered an efficient and inexpensive technique for designing and testing drug candidates [35]. The strong antioxidant, anti-inflammatory, and antidiabetic activities of the phenol-enriched extract of *Causonis trifolia* shoot (PECTS) prompted us to carry out in silico studies to predict the possible mechanism of action of the detected compounds. In structure-based molecular docking, a small ligand molecule is aligned inside the binding cavity of the target protein, and the resulting docking pose is evaluated by a specific scoring function. The scoring function generates a score for each pose, and the resulting values are used to rank the different poses and ligands. In a methodological sense, there are two independent stages in the docking process: pose generation and the scoring. The first refers to the methods that are used to create different ligand and protein conformations and align different ligand conformations within the binding site of the protein. The latter, the scoring, is required in the docking process for the quantitative estimation of the pose quality. In our present study, 19 bioactive phytocompounds were identified from PECTS by GC–MS analysis, and these compounds were used for molecular docking studies. Two anti-inflammatory, one antioxidant, and two antidiabetic target proteins were selected for the molecular docking study. Ergosta-5,22-dien-3-ol, acetate, (3β,22E), was established, employing the iGEMDOCK tool, as the lead molecule among all of the identified compounds, and it exhibited the best anti-inflammatory, antioxidant, and antidiabetic activities by establishing the maximum binding interaction with respective to the target proteins at the cost of the lowest binding energy. When these phytocompounds were subjected to further docking study using Autodock 4.2, they were evaluated based on their free energy of binding (∆G). The free energy of binding (∆G), which is equal to the free energy of the protein–ligand complex minus the free energies of the protein and the ligand in their unbound states, is used to quantitatively measure the degree of spontaneity and strength of protein–ligand binding [55]. The binding interactions study conducted using the Autodock 4.2 programme revealed that the compound ergosta-5,22-dien-3-ol, acetate, (3β,22E), required total binding energy (−11.73 Kcal/mol) with aldo reductase, followed by α-amylase (−9.27 Kcal/mol), Cox-2 (−8.93 Kcal/mol), TNF-α (−8.21 Kcal/mol), and superoxide dismutase (−7.16 Kcal/mol). Enzymes are one of the most important groups of drug targets, and the identification of possible ligand–enzyme interactions is of major importance in many drug discovery processes [56]. Aldo reductase, an enzyme belonging to the aldo keto reductase superfamily, catalyses the rate-limiting step of the polyol pathway, an alternative path for glucose metabolism [57]. Reactive oxygen species (ROS) are produced when glucose is processed through the polyol pathway in hyperglycaemic situations [58]. These metabolic alterations cause osmotic and oxidative stresses, which in turn cause various degenerative diseases [59]. Additionally, the polyol pathway contributes to a number of biochemical alterations, including an increase in the generation of advanced glycation end-products and the activation of protein kinase C, both of which could be relevant to diabetes-induced vascular dysfunction [58]. Since aldo reductase is a central molecule and is known to control the rate-limiting step of the polyol pathway, its inhibition provides a possible strategy for preventing complications of chronic diabetes [60]. Experimental studies suggest that the inhibition of AR could be effective in the prevention of diabetic complications [61]. Thus, identifying potent AR inhibitors can pave the way for effective therapies against diabetes and related complications. In analogy with any spontaneous process, protein–ligand binding occurs only when the change in the Gibbs free energy (ΔG) of the system is negative at a constant pressure and temperature. Because the protein–ligand association extent is determined by the magnitude of the negative ΔG, ergosta-5,22-dien-3-ol, acetate, (3β,22E), could be a very potent inhibitor of aldo reductase. The van der Waals + hydrogen bonding + desolvation energy for ergosta-5,22-dien-3-ol, acetate, (3β,22E), and aldo reductase docked complex was −13.44 kcal/mol, much higher than their electrostatic energy of −0.08 kcal/mol, thus proposing that the hydrogen bonding and van der Waals interaction are the major forces stabilising the protein–ligand complex. Visualisation of the docked complex with the help of pyMol confirmed that the binding of ergosta-5,22-dien-3-ol, acetate, (3β,22E), with aldo reductase being stabilised by four bond interactions involving four amino acid residues (Thr-19, Lys-21, Ser-22, and Tyr-48) of the target protein. Our study recommends further in-depth investigation of this molecule to evaluate its bioactivities and toxicity. The proposed molecule should be subjected to the necessary clinical trials for broad-spectrum drug discovery.

To ensure the safety of plant products, systematic studies are required to predict the risks of toxicity and deliver scientific knowledge regarding the selection of safe dosages of the plant extracts [62]. In our experiment, the PECTS did not show any significant cell death when applied against Vero cells (ATCC-CCL-81) using the MTT assay, even at the highest dose of 250 mg/mL. After the clearance in the in vitro cytotoxicity assay, the PECTS was tested on mice for 14 days in an acute toxicity study. Here, we also did not observe any behavioural abnormalities, as well as deaths, of any mice (including both female and male) at very high doses (5000 mg/kg body weight). Thus, the use of this extract up to the highest administered dose was completely safe for the animals. According to the harmonised system for the classification of chemicals that cause acute toxicity, adopted by the Organization for Economic Co-operation and Development (OECD), this plant extract can strongly be recommended as a nontoxic and safe drug for humans and other animals. However, further repeated long-term chronic toxicological investigations are highly recommended to confirm its safety and effectiveness in humans.

## 4. Materials and Methods

### 4.1. Collection and Preparation of the Plant Materials

The fresh shoots of *Causonis trifolia* (L.) Mabb. & J.Wen were collected from roadside bushes in Santiniketan, Birbhum, West Bengal, India (11°03′15.46″ N, 076°32′23.58″ E) in October 2020. The collected plant species were identified with the help of the *Flora of West Bengal* [63] and was authenticated by an expert in plant taxonomy. The nomenclature of the identified plant species was updated following using a standard website, namely, “Plants of the World Online” [64]. The collected plant specimen was preserved as a herbarium specimen following standard herbarium techniques [65] and kept at the Department of Botany, Visva-Bharati, Santiniketan, India, for future references (voucher specimen number: INDIA, West Bengal, Birbhum district, Santiniketan, 10.10.2020, S Hazra 3 (VBH)). The collected plant materials were washed thoroughly under running tap water to remove the dust particles before they were cut into small pieces, shade dried, and uniformly ground using an electric grinder. The crude powder was stored in an airtight cellophane bag at 4 °C until further use.

### 4.2. Statistical Optimisation of the Extraction Process for the Enhanced Extraction of Phenols

In our present experiment, the response surface methodology (RSM) statistical tool was employed for the optimisation of the extraction of phenols from the shoots of *Causonis trifolia*. A three-factor and three-level Box–Behnken design was applied here for the RSM-based optimisation study (Table 4). A total of 17 experimental runs, including five central points, were generated using Design-expert 13 software (Stat-Ease Inc., Minneapolis, MN, USA) (Table 5). Here, the variables taken for this statistical design were the concentration of the extracted solvent (A, % *v/v*, methanol:water), temperature during the extraction (B, °C), and the duration of the extraction (C, hours). Each variable had two levels: −1 (i.e., lower limit) and +1 (i.e., higher limit). First, four different solvent systems, namely, methanol, ethyl acetate, chloroform, and petroleum ether, were examined to understand their extraction yield of phenols, and methanol was selected, as it exhibited a higher extraction potential. The extraction efficiency of the aqueous methanol was substantially higher than that of 100% methanol, as 100% methanol is not capable of extracting most of the bioactive phytochemicals. Thus, different proportions of methanol and water (ranging from 40% to 100%) (A, % *v/v*, methanol:water) were used as an independent variable for this investigation. The temperature range for this experiment was 20 °C to 80 °C (B, °C), while the extraction time ranged from 1 h to 24 h (C, hours). Meanwhile, the total phenolic content (TPC) was selected as the dependent or response variable. In Table 4, we have listed the ranges of all three parameters considered for the extraction of phenols during the RSM study. Multiple linear regression analysis was performed, and the experimental data were fitted to the second-order polynomial mode.

For further detailed phytochemical analyses and bioactivity studies, the optimised phenol extract of the studied plant part was selected. As per the suggested extraction protocol of the RSM model, the unit mass (10 g) shoot powder was mixed with 150 mL of 67% methanol and kept at 49 °C for 12.53 h under constant agitation in a shaker incubator. The slurry obtained was then filtered through Whatman No. 1 filter paper, and the filtrate was collected. This procedure was repeated thrice to extract all phytochemicals from the powdered plant sample. All obtained filtrates were combined, and the methanol was evaporated in a rotary evaporator machine. The dried extract was stored in an air-tight container at 4 °C until further use.

### 4.3. Quantitative Estimation of Certain Groups of Phytochemicals of the PECTS

For the quantitative estimation of the extract, conventional methods were adopted. The total phenolic content using the Folin- Ciocaletu method [66], the total flavonoid content via the aluminium chloride colorimetric method [67], the total tannin content following the method of Afify et al. (2012) [68], and the total alkaloid content using the 1,10-phenanthroline method [69] were estimated.

### 4.4. High-Performance Liquid Chromatography (HPLC) Analysis of the PECTS

Agilent’s 1260 Infinity II (Santa Clara, CA, USA) HPLC instrument was used, and the data were processed employing OpenLab software. The separation was achieved using a Luna C18 reversed-phase column (25 cm × 4.6 mm, 5 µm) (Phenomenex, Torrance, CA, USA). In our present study, at first, the stock solutions (1 mg/mL) of both the standard compounds and the plant extract were prepared. For this, a 1 mg sample was dissolved in 0.5 mL HPLC-grade methanol, followed by 10 min of sonication, and the resulting volume was 1 mL with the mobile phase solvent (acetonitrile and 1 percent aqueous acetic acid, 1:9). The mobile phase contained 1% aqueous acetic acid solution (Solvent A) and acetonitrile (Solvent B). The flow rate was set at 0.7 mL/min, the column was maintained at 28 °C, and the volume of the injected plant sample was fixed at 20 µL. A gradient elution was performed by changing the proportion of solvent B to solvent A. For a duration of 28 min, the gradient elution was changed from 10% to 40% B in a linear fashion, from 40% to 60% B in 39 min, and from 60% to 90% B in 50 min. It took 55 min for the composition of the mobile phase (solvent B: solvent A) to return to its original state. Then, it was allowed to run for another 10 min before injecting the next sample. The total duration of the analysis per sample was 65 min. According to the absorption maxima of the analysed compounds, the HPLC chromatograms were detected using a photo diode array UV detector at three distinct wavelengths (272, 280, and 310 nm). The retention period of each chemical was then determined by spiking with standards under the same conditions. The integrated peak area was used to quantify the phenolic and flavonoids present in the sample, and the content was estimated using a calibration curve by plotting the peak area against the concentration of the respective standard sample. The data are reported with a convergence limit in triplicate.

### 4.5. Gas Chromatography-Mass Spectrometry (GC–MS) Analysis of the PECTS

Five grams of optimised phenolic extract (PECTS) were dissolved in 100 mL of methanol. The solution was stirred continuously for 72 h before being filtered using Whatman No. 41 filter paper. Then, the extract was completely concentrated by evaporating the methanol in a rotary evaporator at 40 °C. Thereafter, 1 g of the dried concentrated extract was dissolved in 10 mL methanol. The stock solution of the sample extract was then transferred to an airtight container and stored at 4 °C until it was needed for the experiment.

The gas chromatography-mass spectrometry (GC-MS) analysis was carried out on a mass spectrometer (Agilent 6890-MS, Santa Clara, CA, USA). After filtering the sample solution via a sterilised Millipore filter (0.22 µm), it was used for the GC-MS analysis. One microliter of the sample was injected into a GC equipped with an MS and the nonpolar capillary column HP-5 (30 m × 0.25 mm; 0.25 μm). The oven program began at an initial temperature of 110 °C for 2 min, and the transfer line temperature was 200 °C. Then, the temperature was increased to 250 °C for 2 min at a rate of 5 °C/min. Finally, the temperature was raised to 280 °C at a rate of 6 °C/min for 7 min. The total run time was 50 min. The detector temperature was set at 250 °C. Helium (purity: 99.999%) was employed as the carrier gas with a flow rate of 1 mL/min. The sample injection was conducted following the split mode (1:10). In addition, 70 eV was set for the electron impact ionisation (EI). The mass range scanned was 50–550 m/z.

The identification of phytochemicals present in the optimised phenolic extract of *Causonis trifolia* was made by consulting the mass spectral library of NIST (NIST ver. 2.0, 2005). The NIST database was carefully checked, the spectra of the reference compounds were extracted, and they were matched with the unknown phytochemical compounds present in the plant sample.

### 4.6. In Vitro Antioxidant Activity Study of the PECTS

#### 4.6.1. DPPH Radical Scavenging Activity

The DPPH radical scavenging activity was determined following the standard method [70] with slight modifications. The standard curve was prepared with ascorbic acid. The results are expressed as the % of the scavenging activity. The IC_50_ value was determined from the % of the inhibition vs. concentration of the different plant extracts and ascorbic acid by comparing the absorbance values of the control (*A*_0_) and test compounds (*A_t_*). The percentage of the DPPH radical scavenging activity was determined employing the following formula:Radical scavenging activity (%) = (*A*_0_ − *A*_*t*_/*A*_0_) × 100(1)

#### 4.6.2. Hydrogen Peroxide Scavenging Assay

The ability of the optimised phenolic extract of *C. trifolia* shoots to scavenge hydrogen peroxide was determined according to the method of Ruch et al. (1989) [71]. The IC_50_ value was determined from the % of the inhibition vs. concentration of the plant extract and ascorbic acid by comparing the absorbance values of the control (*A*_0_) and test compounds (*A_t_*). The percentage of the H_2_0_2_ radical scavenging activity was determined using the following formula:Radical scavenging activity (%) = (*A*_0_ − *A_t_*/*A*_0_) × 100(2)

#### 4.6.3. ABTS Radical Scavenging Activity

This assay was carried out following the standard protocol of Thaipong et al. (2006) [70]. The standard curve was prepared using ascorbic acid. The results are expressed as the % of the scavenging activity. The IC_50_ value was determined from the % of the inhibition vs. concentration of the different plant extracts and ascorbic acid by comparing the absorbance values of the control (*A*_0_) and test compounds (*A*_t_). The percentage of the ABTS radical scavenging activity was determined using the following formula:Radical scavenging activity (%) = (*A*_0_ − *A_t_*/*A*_0_) × 100(3)

#### 4.6.4. Phosphomolybdenum Assay

The total antioxidant activity of the extracts was evaluated using the phosphomolybdenum method [72]. Each test tube contained 0.3 mL of the optimised extract which was mixed with 3 mL of reagent solution (0.6 M sulfuric acid, 28 mM sodium phosphate, and 4 mM ammonium molybdate) and then incubated at 95 °C for 90 min. After the incubation, the absorbance of the test solution was measured at 695 nm using a UV-Vis spectrophotometer (Shimadzu UV1800 double beam spectrophotometer, Toshvin Analytical Pvt. Ltd., Kytoto, Japan) against a blank after cooling at room temperature. The total antioxidant activity is expressed as the milligram equivalent of ascorbic acid per gram sample.

### 4.7. In Vitro Anti-Inflammatory Activity Study of the PECTS

#### 4.7.1. Inhibition of Albumin Denaturation Assay

First, 2 mL of 1% bovine serum albumin (BSA) was mixed with 400 μL of plant extract at different concentrations (100–500 μg/mL), and the pH of the reaction mixture was adjusted to 6.8 with 1N HCl. The reaction mixture was incubated at room temperature for 20 min and then heated to 55 °C for 20 min in a water bath. The mixture was cooled to room temperature, and the absorbance value was recorded at 660 nm. A BSA mixture with a 30% methanol solution was used as the control. An aqueous solution of diclofenac sodium at different concentrations was used as the standard [73]. The experiment was performed in triplicate. The percent inhibition was calculated using the following formula:(4)$$\mathrm{Inhibition}\;\left(\%\right)=\frac{\mathrm{Control}\;O.D.-\;\mathrm{Sample}\;O.D.}{\mathrm{Control}\;O.D.}\times 100$$
where Control O.D. = optical density of the control; Sample O.D. = optical density of the test sample.

#### 4.7.2. Antiprotease Activity Assay

First, 1 mL of 20 mM Tris HCl buffer (pH 7.4) was mixed with 0.06 mg of trypsin and 1 mL of extract at different concentrations (100–500 μg/mL). The reaction mixture was incubated at room temperature (37 °C) for 5 min. Then, 0.8% casein was added to the reaction mixture and further incubated for 20 additional minutes. A total of 2 mL of 70% perchloric acid was added to the mixture to terminate the reaction. The solution was centrifuged at 3000 rpm for 10 min. The absorbance of the supernatant was recorded at 210 nm. An aqueous solution of diclofenac sodium at different concentrations was used as the standard. The IC_50_ value was calculated [73]. The experiment was performed in triplicate. The percent inhibition was calculated using the following formula:(5)$$\mathrm{Inhibition}\;\left(\%\right)=\frac{\mathrm{Control}\;O.D.-\;\mathrm{Sample}\;O.D.}{\mathrm{Control}\;O.D.}\times 100$$

#### 4.7.3. Membrane Stabilisation Assay

##### A. Preparation of the R.B.C. Suspension

A blood sample was centrifuged at 3000 rpm for 10 min and washed thrice with normal isotonic saline. Then, 10% (*v/v*) normal saline was added to the tubes and stored at 4 °C for further experimental use.

##### B. Heat-Induced Haemolysis Assay

First, 1 mL of 10% (*v/v*) RBC suspension was mixed with 1 mL of the plant extract (100–500 μg/mL). The reaction mixture was incubated in a water bath at 56 °C for 30 min. The tubes were cooled to room temperature under running tap water. The tubes were centrifuged at 2500 rpm for 5 min, and the absorbance value of the supernatant was determined spectrophotometrically at 560 nm. Diclofenac sodium was used as a standard [73]. The percentage inhibition of the haemolysis was calculated as follows:(6)$$\mathrm{Inhibition}\;\left(\%\right)=\frac{\mathrm{Control}\;O.D.-\;\mathrm{Sample}\;O.D.}{\mathrm{Control}\;O.D.}\times 100$$

### 4.8. In Vitro Antidiabetic Activity Study of the PECTS

#### 4.8.1. α-Amylase Inhibitory Assay

The assay was carried out following the standard protocol [74]. Starch azure (2 mg) was suspended in 0.2 mL of 0.5 M Tris–HCl buffer (pH 6.9) containing 0.01 M CaCl_2_ (i.e., substrate solution). The tubes containing the substrate solution were boiled for 5 min and then preincubated at 37 °C for 5 min. The plant extracts were dissolved in DMSO in order to obtain concentrations of 50, 100, 150, and 200 μg/mL. Then, 0.2 mL of plant extract of particular concentration was added to the tube containing the substrate solution. In addition, 0.1 mL of porcine pancreatic amylase in Tris–HCl buffer (2 units/mL) was added to the tube containing the plant extract and substrate solution. The reaction was carried out at 37 °C for 10 min, and it was stopped by adding 0.5 mL of 50% acetic acid to each tube. The reaction mixture was centrifuged at 3000 rpm for 5 min at 4 °C. The absorbance of the resulting supernatant was measured at 595 nm using a spectrophotometer (Shimadzu UV1800 double beam). Acarbose, a known α-amylase inhibitor, was used as the standard drug. The experiments were repeated thrice. The percentage of the α-amylase inhibitory activity was calculated using the following formula:(7)$$\mathrm{Inhibition}\;\left(\%\right)=\frac{\mathrm{Control}\;O.D.-\;\mathrm{Sample}\;O.D.}{\mathrm{Control}\;O.D.}\times 100$$

The α-amylase inhibitory activities of the plant extracts and acarbose were calculated, and their IC_50_ values were determined.

#### 4.8.2. Glucose Uptake Assay Using Yeast Cells

A glucose uptake assay was performed using yeast cells according to the method of Cirillo et al. (1963) [75]. Commercial baker’s yeast suspended in distilled water was subjected to repeated centrifugation (3000 rpm for 5 min) until a clear supernatant was obtained, and 10% (*v/v*) of the suspension was prepared in distilled water. Various concentrations (50, 100, 150, and 200 μg/mL) of the plant extract were added to 1 mL of glucose solution (5 mM) and incubated together for 10 min at 37 °C. The reaction was started by adding 100 μL of the yeast suspension followed by vortexing and further incubation at 37 °C for 60 min. After the incubation, the tubes were centrifuged for 5 min at 3800 rpm, and the glucose was estimated using a spectrophotometer (Shimadzu UV1800 double-beam spectrophotometer) at 520 nm. Acarbose was used as the standard drug. All experiments were carried out in triplicate. The percentage increase in the glucose uptake by the yeast cells was calculated using the following formula:(8)$$\mathrm{Glucose}\;\mathrm{uptake}\;\left(\%\right)=\frac{\mathrm{Control}\;O.D.-\;\mathrm{Sample}\;O.D.}{\mathrm{Control}\;O.D.}\times 100$$

### 4.9. In Silico Molecular Docking Study

A docking study was performed to predict the possible anti-inflammatory, antioxidant, and antidiabetic properties of the phytocompounds identified from the GC-MS analysis of the shoot extract of the studied plant. The chemical structures of the compounds were retrieved as SDF files from the PubChem database “https:/pubchem.ncbi.nlm.nih.gov (accessed on 1 September 2021)”. Then, the downloaded SDF files were converted into Mol, PDBQT, and PDB file formats using OPEN BABEL software “https://openbabel.org/wiki/Main Page (accessed on 1 September 2021)”. The three-dimensional (3D) structures of the phytocompounds were optimised for molecular docking using Chimera 1.15 software [76]. Here, we employed a structure-based molecular docking approach to study the interactions of the phytocompounds with the proteins responsible for triggering inflammatory (e.g., tumour necrosis factor alpha (Tnf α) (PDB ID: 2AZ5) and cyclooxygenase-2 (Cox2) (PDB ID: 4PH9)); oxidative stress (superoxide dismutase (SOD) (PDB ID: 5YTO)); and diabetic (e.g., α-amylase (PDB ID: 1B2Y) and aldo reductase (PDB id: 4YS1)) conditions in the human body. The Research Collaboratory for Structural Bioinformatics (RCSB) Protein Data Bank’s “https://www.rscb.org (accessed on 2 September 2021)” online server was exploited to obtain the structures of the selected target proteins. Finally, the molecular docking experiments were carried out using the iGEMDOCK 2.1 and AutoDock 4.2 programs [77,78].

Initially, hydrogen bonds, bond orders, charges, and flexible torsions were assigned to all of the target proteins and phytocompounds in order to prepare them for the docking study. The iGEMDOCK application was used to screen these compounds for protein–ligand interactions. The predefined parameters (population size: 200, generations: 60, and solutions: 2) were used in this docking study. The conformation with the lowest total binding energy among the different conformations generated was considered the best binding conformation of the phytocompounds against the target proteins. The identified phytocompounds were imported into the iGEMDOCK graphical user interface and sorted by the postdocking analysis based on their binding energies and compound fitness score measured using the iGEMDOCK docking algorithm [35]. The most potent phytocompound against the five target proteins was subjected to a further docking study using the Autodock 4.2 programme. Autodock 4.2 was employed to screen for the optimal binding position of the ligands along with an assessment of the relative intensities of the binding interactions. The interacting protein residues forming hydrogen bonds with phytocompounds were visible in the Pymol viewer [79].

### 4.10. Toxicity Analysis of the PECTS

#### 4.10.1. In Vitro Cytotoxicity Assay

A cell toxicity assay was performed against the Vero cell line (ATCC-CCL-81) using the 3-(4,5-dimethylthiazol-2-yl)-2,5-diphenyltetrazolium bromide (MTT) assay following the standard protocol [80]. Approximately 5 × 10^3^ cells/well were seeded in a 96-well plate and incubated at 37 °C with a 5% CO_2_ atmospheric condition. After 24 h, the PECTS were added at concentrations ranging from 50 to 250 mg/L, and the incubation was conducted for 72 h. After the incubation, MTT was added at a 5 mg/L concentration to each well and the incubation was continued at 37 °C for a further 4 h. The residual medium was discarded, 0.1 mL of DMSO was added to solubilise the formazan crystals, and OD was taken at 540 nm for calculating the CC_50_. The lowest concentration of a compound that resulted in a 50% reduction in the cell viability is called CC_50_. The positive control was doxorubicin, and each experiment was repeated three times. The percentage of the cell viability was calculated using the following formula:(9)$$\mathrm{Cell}\;\mathrm{viability}\;\left(\%\right)=\frac{\mathrm{Absorbence}\;\mathrm{of}\;\mathrm{treated}\;\mathrm{cells}\;-\mathrm{Background}\;\mathrm{absorbence}}{\mathrm{Absorbence}\;\mathrm{of}\;\mathrm{untreated}\;\mathrm{cells}-\mathrm{background}\;\mathrm{absorbence}}\times 100$$

#### 4.10.2. In Vivo Acute Toxicity Study

##### Animals and Ethical Use

An acute toxicity study was conducted on 6- to 8-week-old BALB/c mice acquired from the National Laboratory Animal Facility of the CSIR-Central Drug Research Institute, Lucknow. During the experiment, the Animal Ethical Committee’s standards were strictly observed. Four experimental groups, each with 6 mice, were formed from a total of 24 male and female BALB/c mice weighing 20–22 g (3 males and 3 females per group). The animals were acclimatised for 1 week, maintained in standard laboratory conditions (24 ± 2 °C) with 12:12 h light and dark cycles in individual polypropylene cages, and they were fed with a pelleted standard rat diet (Lipton India Ltd., Bangalore, India) and water ad libitum. The experimental protocol and report were approved by the Institutional Animal Ethical Committee (IAEC) (IAEC/2019/114/Renew-2/Dated 25 May 2022).

##### Experiment Details

An acute toxicity assay was performed according to the Organization for Economic Cooperation and Development’s (OECD) guideline 423 [81]. The acute toxicity study was performed on 6- to 8-week-old BALB/c. During the experiment, the Animal Ethical Committee’s standards were strictly observed. A total of 24 male and female BALB/c mice weighing 20–22 g were randomly assigned to four experimental groups, each with six mice (3 males and 3 females per group). After fasting overnight, each treatment group was administered with PECTS at single doses of 1000, 2500, and 5000 mg/kg by oral gavage. The control groups received the same amount of distilled water. The mice were allowed free access to feed and drinking water. After dosing, all animals were observed individually for mortality and changes in general behaviour during the first 30 min and then at 2, 4, 6, 10, and 24 h following the treatment. Symptoms of toxicity, such as hypo-activity, breathing difficulty, tremors, piloerection, and convulsion, were observed after the administration of the various doses of the extract. During the remaining experimental period, animal observations were conducted at least once a day for a postdosing period of 14 days. The body weights were measured at the initiation of the treatment and on days 4, 7, 11, and 14 after the administration of the extract.

## 5. Conclusions

The present study explored the optimised phenolic extract of *Causonis trifolia* shoot regarding phytochemical analysis and biological activities. This extraction procedure was established using the response surface methodology (RSM), and it was employed as an effective tool for the extraction of the maximum amount of total phenolics from the minimum quantities of shoot biomass of *C. trifolia*. This technique is found to be essential for various purposes, such as extracting the highest number of intact, undistorted, and bioactive phenolic molecules in a cost-effective and rapid manner and facilitating the appearance of higher levels of antioxidant, anti-inflammatory, and antidiabetic properties by the extracted bioactive phenolics. The optimised extract was further analysed by HPLC and GC-MS analyses to identify and quantify the various bioactive compounds. An impressive and effective phytochemical fingerprint was explored in our study, which will further be employed to highlight the immense potential of the studied herb while curing various diseases and will also serve as a characteristic identifier of the studied plant extract. We have also experimentally demonstrated the antioxidant, anti-inflammatory, and antidiabetic potentials of the optimised phenolic shoot extract of the investigated plant. Although we focused on the bioactivities of the optimised phenolic extract, this study was not sufficient to detect specific active principle(s). In future various scientific projects, this can be undertaken to further fractionate this highly potential nexus of polyphenols. Despite the fact that the in vitro cell toxicity assay against the Vero cell line and the in vivo acute toxicity study on BALB/c mice showed that this extract is nontoxic, still, a detailed subacute and chronic toxicity study is required to comprehensively characterise the toxicity profile of this extract and to certify the hazardless use of the studied plant. From an in silico molecular docking study, we predicted that ergosta-5,22-dien-3-ol, acetate, (3β,22E), compound has the most promising binding affinity toward the binding sites of different target proteins associated with oxidative, inflammatory, and diabetic conditions in the human body. In the future, the identified phytocompound can be validated through molecular dynamics simulations of the protein model and experimental research on animal models to confirm its status as a novel compound. Our study extends the scope of scientific studies on this unexplored ethnomedicinal plant for the validation of its ethnomedicinal claims and development of a nontoxic drug candidate with maximum potential.

## Figures and Tables

**Figure 1 plants-12-01495-f001:**
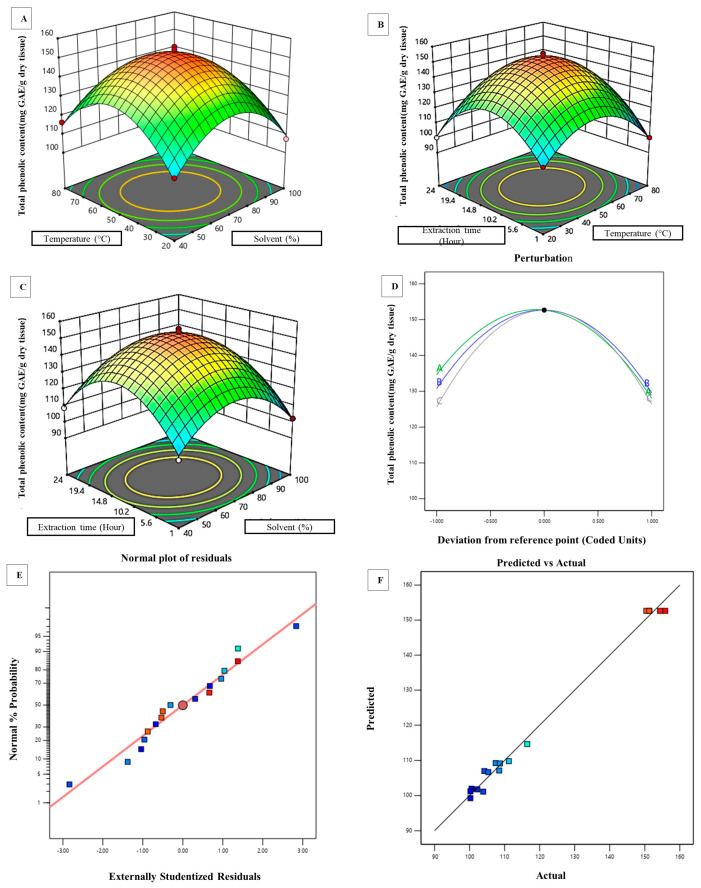
Statistical optimisation of the extraction parameters by the response surface methodology. The response surface plots demonstrate the combined effect of the (**A**) temperature (°C) and solvent (%); (**B**) extraction time (hour) and temperature (°C); (**C**) extraction time (hour) and solvent (%) on the yield of TPC. (**D**) Perturbation plot showing the relative influence of each parameter on the extraction of TPC. (**E**) Normal probability plot showing scatters along the standard curve supporting the prediction of the ANOVA analysis. (**F**) Comparison of the experimental and predicted results, showing a greater degree of agreement between them.

**Figure 2 plants-12-01495-f002:**
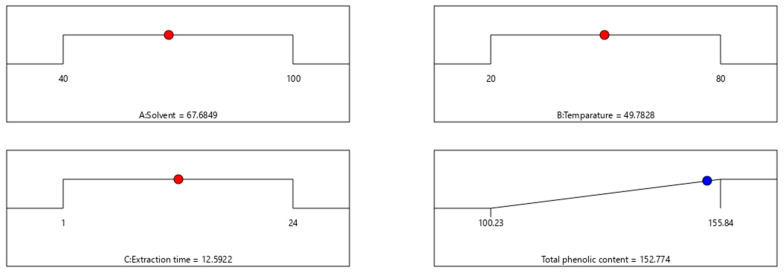
Predictive extraction model designed for the effective extraction of the highest amount of bioactive phenols. The model highlights the most suitable values of the three parameters, namely, methanol concentration (67.6754%), extraction temperature (49.7753 °C), and extraction time, (12.5837 h) predicted for the highest yield of bioactive phenols (152.774 mg GAE/g dry tissue).

**Figure 3 plants-12-01495-f003:**
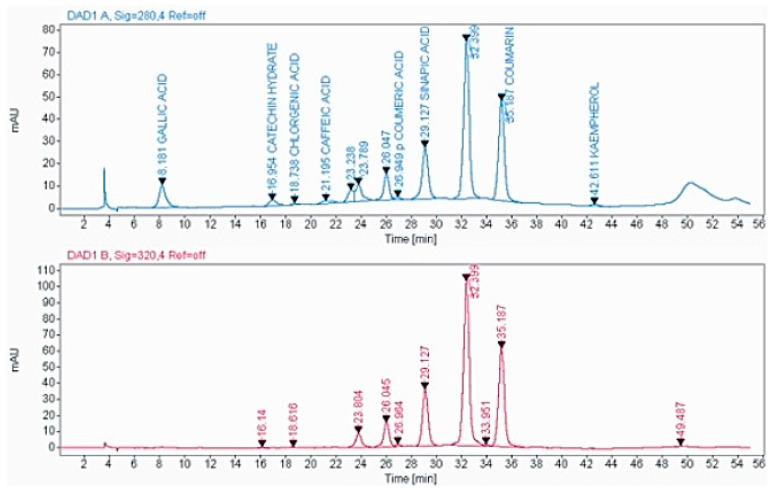
HPLC chromatogram obtained from the optimised phenolic extract of *Causonis trifolia* shoot (PECTS).

**Figure 4 plants-12-01495-f004:**
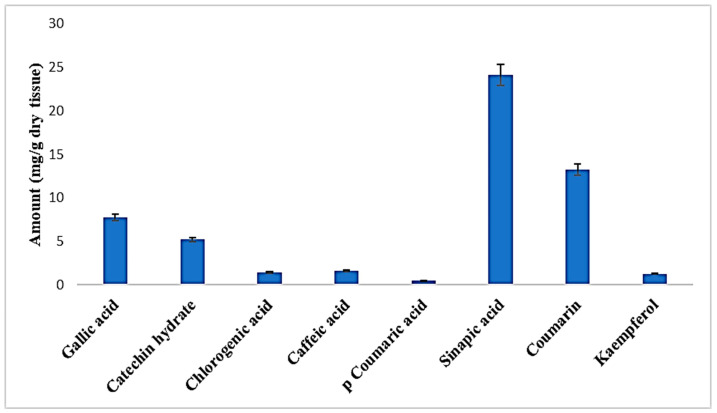
Phenolic and flavonoid profiles of the optimised phenolic extract of *Causonis trifolia* shoot (PECTS).

**Figure 5 plants-12-01495-f005:**
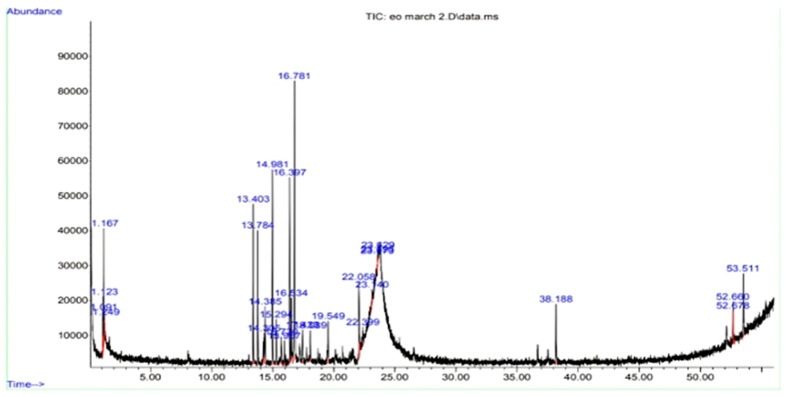
GC-MS chromatogram obtained from optimised phenolic extract of *Causonis trifolia* shoot.

**Figure 6 plants-12-01495-f006:**
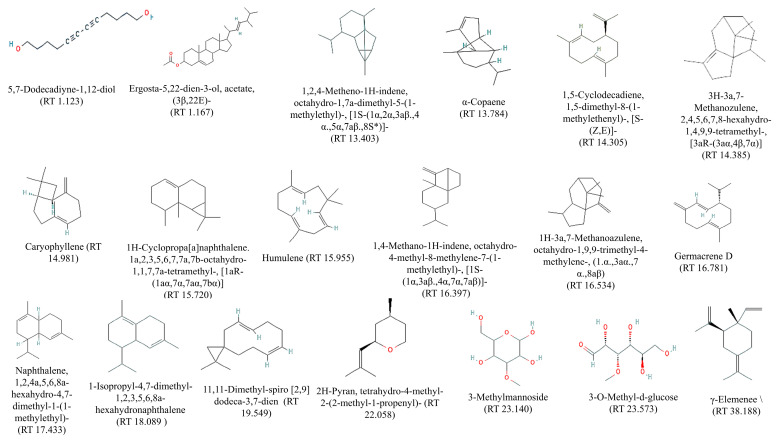
Chemical structure and retention time of the compounds identified from PECTS.

**Figure 7 plants-12-01495-f007:**
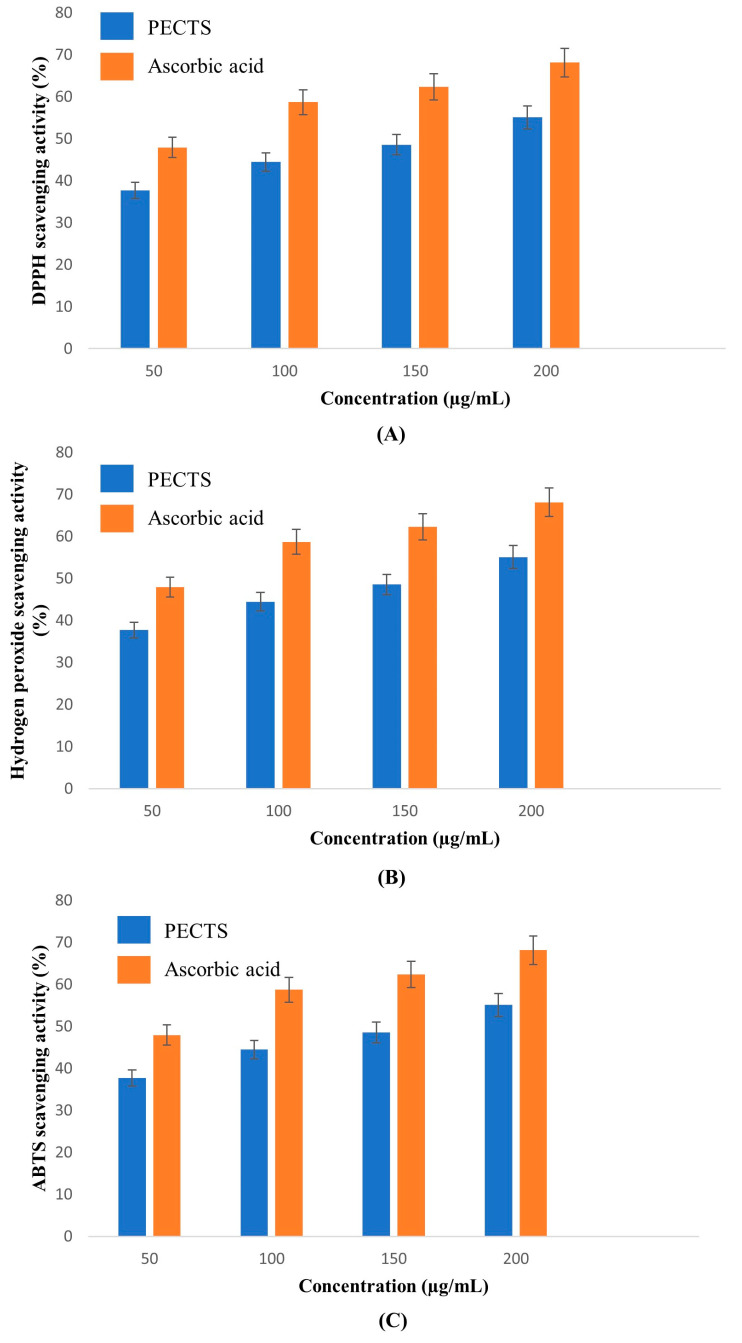
In vitro antioxidant activities exhibited by the PECTS in different radical scavenging assays and a comparison with ascorbic acid: (**A**) DPPH radical scavenging activity of PECTS (Y = 0.5907x + 14.557; IC_50_ = 59.96 μg/mL) and ascorbic acid (Y = 0.4368x + 37.336; IC_50_ = 28.99 μg/mL); (**B**) hydrogen peroxide scavenging activity of the PECTS (Y = 0.5288x + 13.969; IC_50_ = 68.13 μg/mL) and ascorbic acid (Y = 0.4526x + 33.847; IC_50_ = 35.68 μg/mL); (**C**) ABTS radical scavenging activity of the PECTS (Y = 0.58x − 2.0551; IC_50_ = 47.94 μg/mL) and ascorbic acid (Y = 0.4406x + 35.233; IC_50_ = 33.51 μg/mL).

**Figure 8 plants-12-01495-f008:**
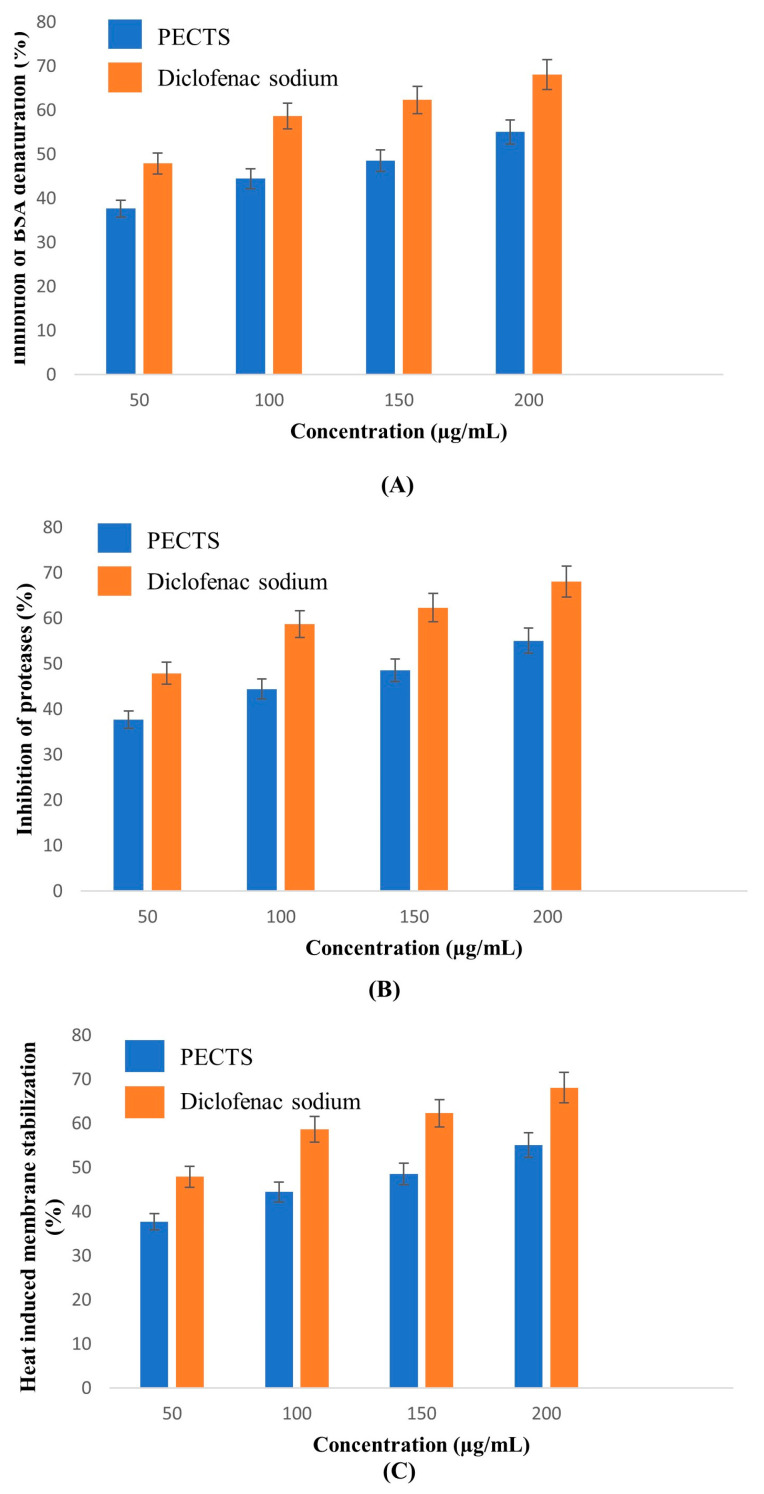
In vitro anti-inflammatory activities of the optimised phenolic extract of *Causonis trifolia* shoot and a comparison with the anti-inflammatory activity of the standard drug diclofenac sodium: (**A**) inhibition of the BSA denaturation activity of th ePECTS (Y = 0.1833x − 6.133; IC_50_ = 306.78 μg/mL) and diclofenac sodium (Y = 0.1852x − 3.66; IC_50_ = 288.74 μg/mL); (**B**) antiprotease activity of the PECTS (Y = 0.1541x + 5.161; IC_50_ = 290.97 μg/mL) and diclofenac sodium (Y = 0.1764x + 2.337; IC_50_ = 270.19 μg/mL); (**C**) inhibition of the heat-induced haemolysis activity of the PECTS (Y = 0.1547x − 0.594; IC_50_ = 327.04 μg/mL) and diclofenac sodium (Y = 0.177x + 5.131; IC_50_ = 253.48 μg/mL).

**Figure 9 plants-12-01495-f009:**
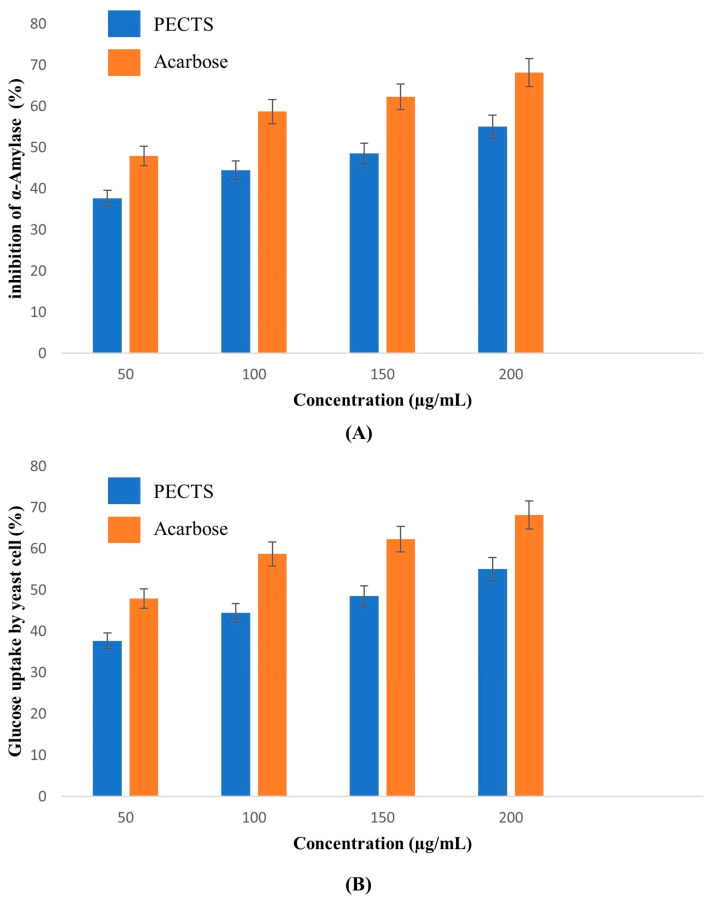
In vitro antidiabetic activity of the optimised phenolic extract of *Causonis trifolia* (PECTS) and a comparison with acarbose’s activity: (**A**) inhibition of the α-amylase activity of the PECTS (Y = 0.3021x − 14.55; IC_50_ = 117.38 μg/mL) and acarbose (Y = 0.2541x − 28.76; IC_50_ = 84.48 μg/mL); (**B**) yeast cell glucose uptake activity of the PECTS (Y = 0.1126x + 32.385; IC_50_ = 156.42 μg/mL) and acarbose (Y = 0.1284x + 43.2; IC_50_ = 52.95 μg/mL).

**Figure 10 plants-12-01495-f010:**
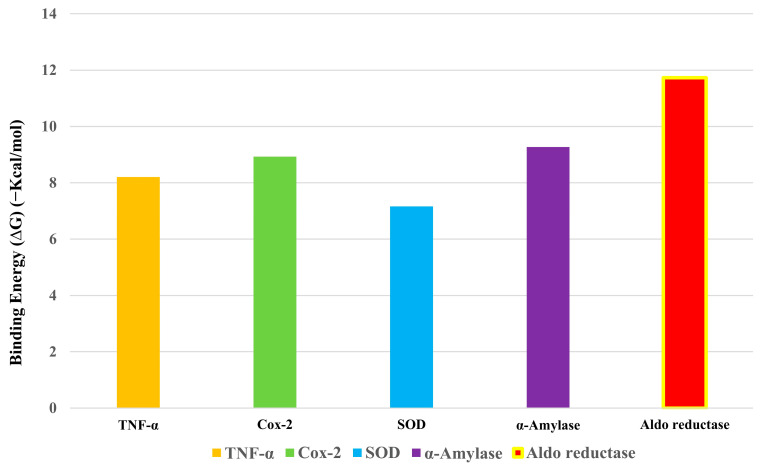
Binding energy (−Kcal/mol) of Ergosta-5,22-dien-3-ol, acetate, (3β,22E) with five target proteins.

**Figure 11 plants-12-01495-f011:**
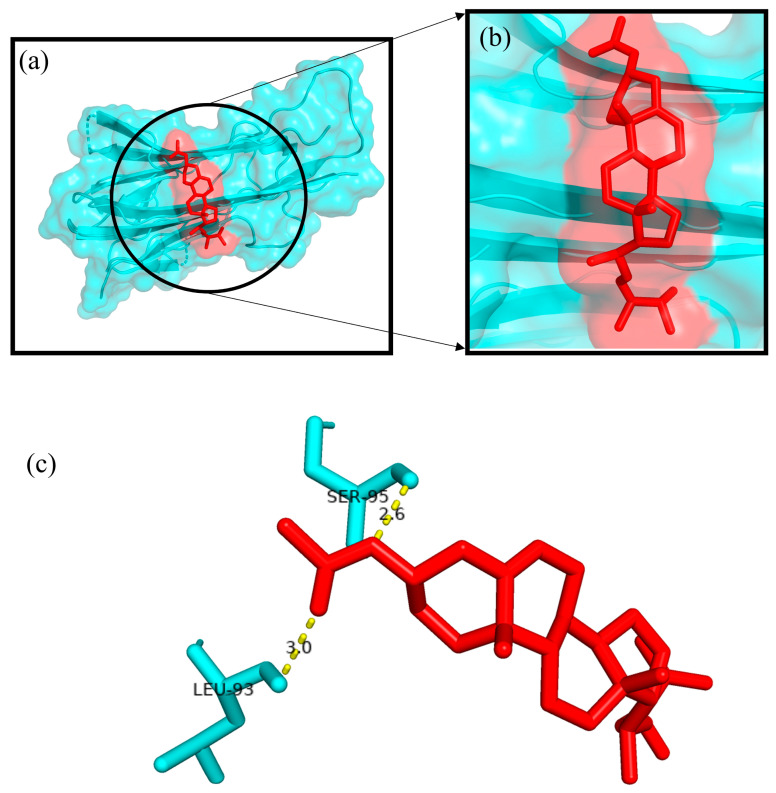
(**a**) Surface view of the docked complex of ergosta-5,22-dien-3-ol, acetate, (3β,22E), and an anti-inflammatory target protein (TNF-α); (**b**) binding pose of ergosta-5,22-dien-3-ol, acetate, (3β,22E); (**c**) ligand–protein docked complex showing the binding interaction of ergosta-5,22-dien-3-ol, acetate, (3β,22E), with TNF-α. Two hydrogen bonds (yellow, dotted line) formed between the ligand ergosta-5,22-dien-3-ol, acetate, (3β,22E), and two amino acid residues (Leu-93 (3.0 Å) and Ser-95 (2.6 Å)) of the target protein TNF-α.

**Figure 12 plants-12-01495-f012:**
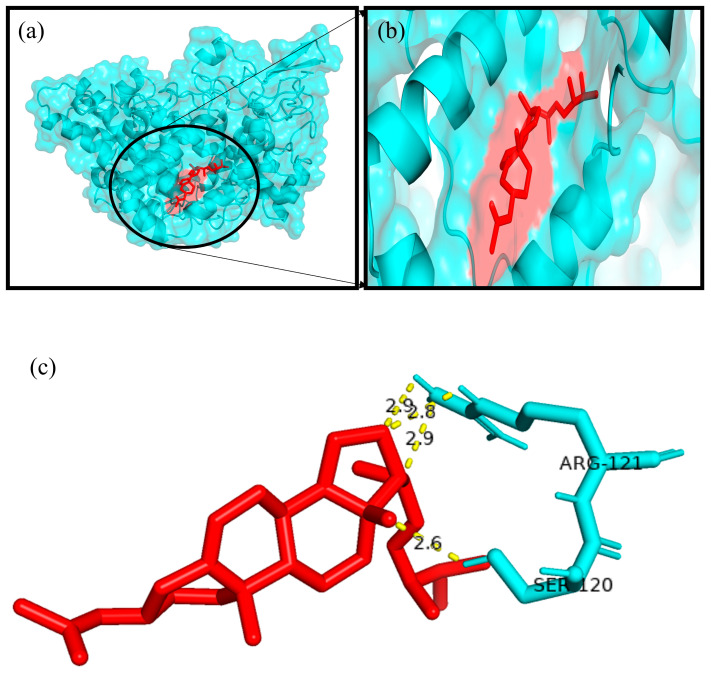
(**a**) Surface view of the docked complex of ergosta-5,22-dien-3-ol, acetate, (3β,22E), and the anti-inflammatory target protein (Cox-2); (**b**) binding pose of ergosta-5,22-dien-3-ol, acetate, (3β,22E); (**c**) ligand–protein docked complex showing the binding interaction of ergosta-5,22-dien-3-ol, acetate, (3β,22E), with Cox-2. Four hydrogen bonds (yellow, dotted line) formed between the ligand ergosta-5,22-dien-3-ol, acetate, (3β,22E), and two amino acid residues (Ser-120 and Arg-121) of the target protein Cox-2.

**Figure 13 plants-12-01495-f013:**
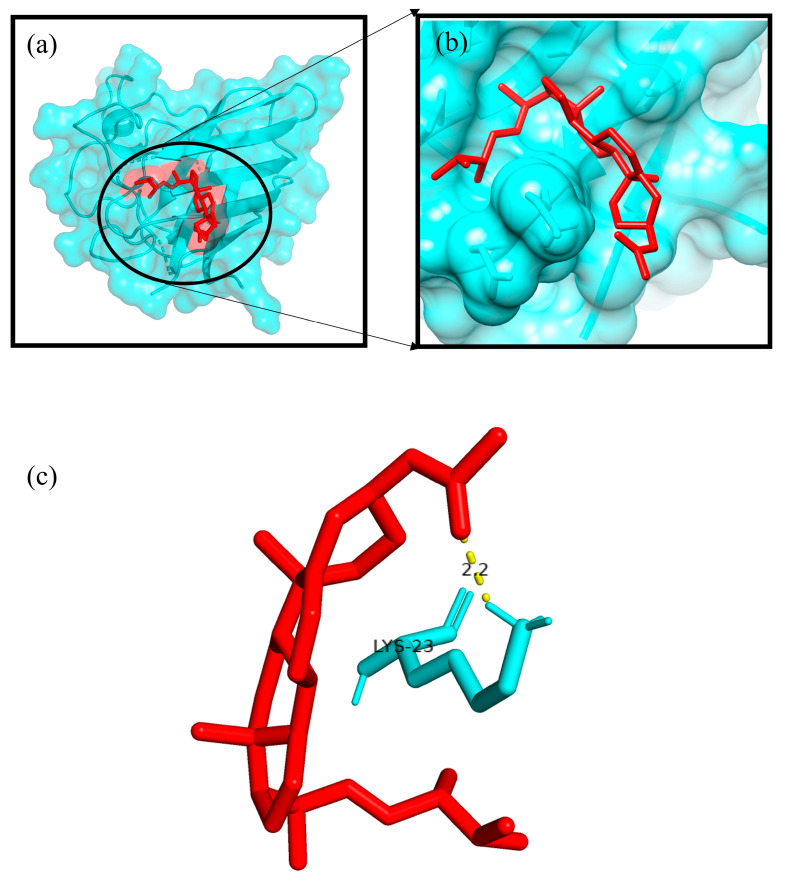
(**a**) Surface view of the docked complex of ergosta-5,22-dien-3-ol, acetate, (3β,22E), and the antioxidant target protein (SOD); (**b**) binding pose of ergosta-5,22-dien-3-ol, acetate, (3β,22E); (**c**) ligand–protein docked complex showing the binding interaction of ergosta-5,22-dien-3-ol, acetate, (3β,22E), with SOD. One hydrogen bond (yellow, dotted line) formed between the ligand ergosta-5,22-dien-3-ol, acetate, (3β,22E), and one amino acid residue (Lys-23 (2.2 Å)) of the target protein SOD.

**Figure 14 plants-12-01495-f014:**
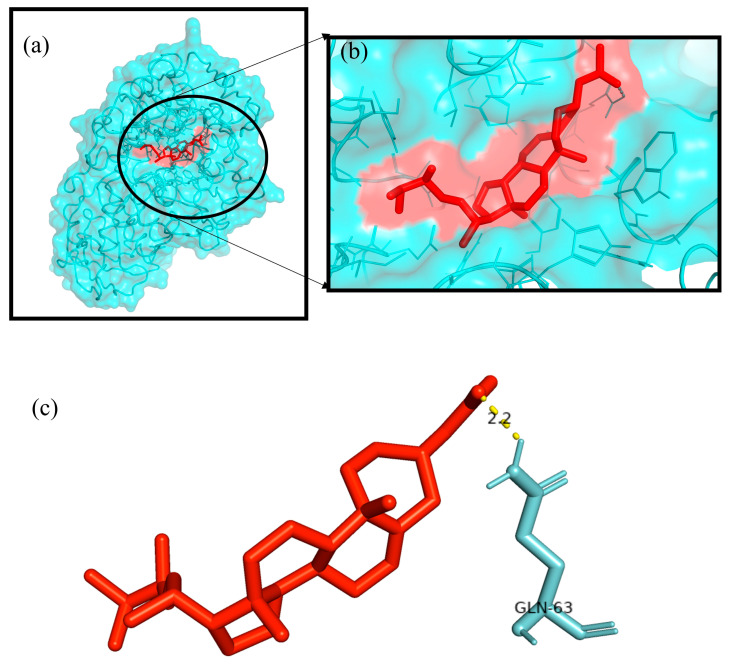
(**a**) Surface view of the docked complex of ergosta-5,22-dien-3-ol, acetate, (3β,22E), and the antidiabetic target protein (α-amylase); (**b**) binding pose of ergosta-5,22-dien-3-ol, acetate, (3β,22E); (**c**) ligand–protein docked complex showing the binding interaction of ergosta-5,22-dien-3-ol, acetate, (3β,22E), with α-amylase. One hydrogen bond (yellow, dotted line) formed between the ligand ergosta-5,22-dien-3-ol, acetate, (3β,22E), and one amino acid residue (Gln-63 (2.2 Å)) of the target protein α-amylase.

**Figure 15 plants-12-01495-f015:**
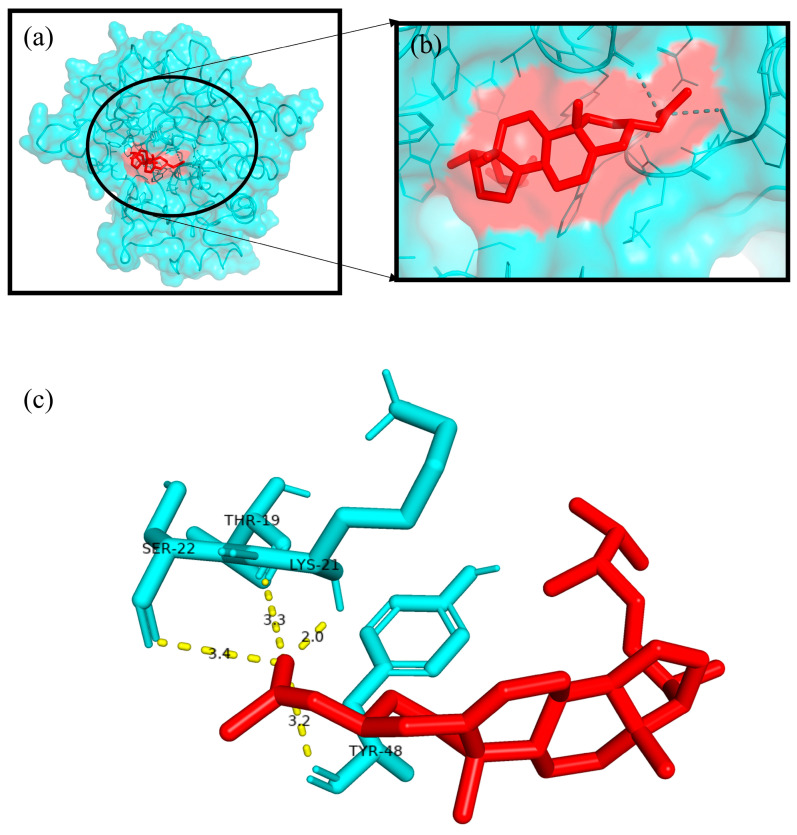
(**a**) Surface view of the docked complex of ergosta-5,22-dien-3-ol, acetate, (3β,22E), and the antidiabetic target protein (aldo reductase); (**b**) binding pose of ergosta-5,22-dien-3-ol, acetate, (3β,22E); (**c**) ligand–protein docked complex showing the binding interaction of ergosta-5,22-dien-3-ol, acetate, (3β,22E), with aldo reductase. Four hydrogen bonds (yellow, dotted line) formed between the ligand ergosta-5,22-dien-3-ol, acetate, (3β,22E), and four amino acid residues (Thr-19 (2.2 Å), Lys-21 (2.0 Å), Ser-22 (3.4 Å), and Tyr-48 (3.2 Å)) of the target protein aldo reductase.

**Figure 16 plants-12-01495-f016:**
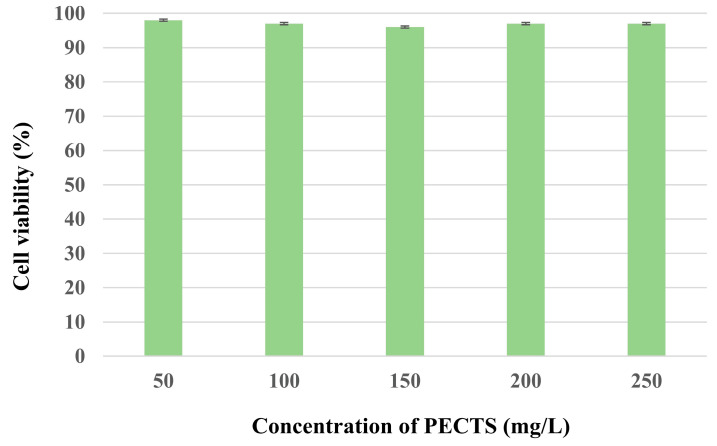
Cell viability (%) at different concentrations (50–250 mg/L) of the PECTS.

**Figure 17 plants-12-01495-f017:**
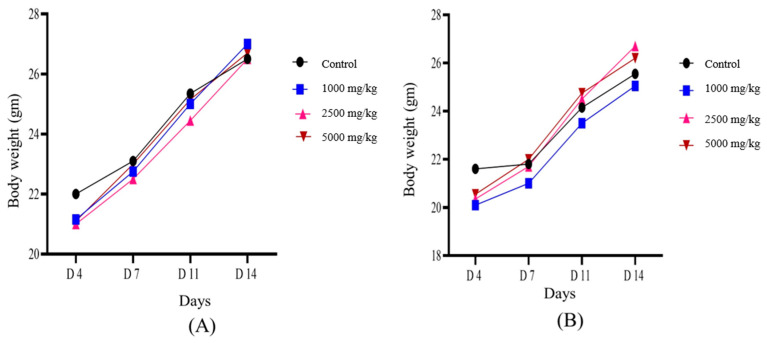
Changes in body weight of (**A**) female BALB/c and (**B**) male BALB/c mice over 14 days of acute exposure (oral administration) of the PECTS.

**Table 1 plants-12-01495-t001:** Phytochemical constituents (*n* = 19) identified in the optimised phenolic extract of *Causonis trifolia* shoot using GC-MS.

Sl. No.	Retention Time (Minute)	PubChem CID	Compound	Molecular Formula	Molecular Weight	Peak Area (%)
1	1.123	560878	5,7-Dodecadiyne-1,12-diol	C_12_H_18_O_2_	194.13	0.979
2	1.167	5352877	Ergosta-5,22-dien-3-ol, acetate, (3β,22E)-	C_30_H_48_O_2_	440.36	5.132
3	13.403	519960	1,2,4-Metheno-1H-indene, octahydro-1,7a-dimethyl-5-(1-methylethyl)-, [1S-(1α,2α,3aβ.,4 α.,5α,7aβ.,8S*)]-	C_15_H_24_	204.18	8.493
4	13.784	19725	α-Copaene	C_15_H_24_	204.18	7.422
5	14.305	20839485	1,5-Cyclodecadiene, 1,5-dimethyl-8-(1-methylethenyl)-, [S-(Z,E)]-	C_15_H_24_	204.18	1.508
6	14.385	99856	3H-3a,7-Methanozulene, 2,4,5,6,7,8-hexahydro-1,4,9,9-tetramethyl-, [3aR-(3aα,4β,7α)]	C_15_H_24_	204.18	2.771
7	14.981	5281515	Caryophyllene	C_15_H_24_	204.18	10.962
8	15.720	28481	1H-Cyclopropa[a]naphthalene. 1a,2,3,5,6,7,7a,7b-octahydro-1,1,7,7a-tetramethyl-, [1aR-(1aα,7α,7aα,7bα)]	C_15_H_24_	204.18	1.478
9	15.955	5281520	Humulene	C_15_H_24_	204.18	1.268
10	16.397	530427	1,4-Methano-1H-indene, octahydro-4-methyl-8-methylene-7-(1-methylethyl)-, [1S-(1α,3aβ.,4α,7α,7aβ)]-	C_15_H_24_	204.18	10.288
11	16.534	521302	1H-3a,7-Methanoazulene, octahydro-1,9,9-trimethyl-4-methylene-, (1.α.,3aα.,7 α.,8aβ)	C_15_H_24_	204.18	3.045
12	16.781	5317570	Germacrene D	C_15_H_24_	204.18	16.755
13	17.433	101708	Naphthalene, 1,2,4a,5,6,8a-hexahydro-4,7-dimethyl-1-(1-methylethyl)-	C_15_H_24_	204.18	2.023
14	18.089	10223	1-Isopropyl-4,7-dimethyl-1,2,3,5,6,8a-hexahydronaphthalene	C_15_H_24_	204.18	1.505
15	19.549	5367333	11,11-Dimethyl-spiro [2,9] dodeca-3,7-dien	C_14_H_22_	190.17	2.921
16	22.058	6432154	2H-Pyran, tetrahydro-4-methyl-2-(2-methyl-1-propenyl)-	C_10_H_18_O	154.13	5.438
17	23.140	247323	3-Methylmannoside	C_7_H_14_O_6_	194.07	0.870
18	23.573	8973	3-O-Methyl-d-glucose	C_7_H_14_O_6_	194.07	0.286
19	38.188	6432312	γ-Elemene	C_15_H_24_	204.18	3.857

**Table 2 plants-12-01495-t002:** Chemical group, structure, and reported biological activity of the phytocompounds identified from the PECTS by GC-MS analysis.

Sl. No.	Compound	Chemical Group	Chemical Structure	Biological Activity	Reference
1	1,2,4-Metheno-1H-indene, octahydro-1,7a-dimethyl-5-(1-methylethyl)-, [1S-(1α,2α,3aβ.,4 α.,5α,7aβ.,8S*)]-	Sesquiterpene	CC(C)C1CCC2(C3C1C4C2(C4C3)C)C	Antiproliferative, genotoxic, and oxidant activities	[21]
2	1,4-Methano-1H-indene, octahydro-4-methyl-8-methylene-7-(1-methylethyl)-, [1S-(1α,3aβ.,4α,7α,7aβ)]	Sesquiterpene	CC©C1CCC2(C3C1C(C2=C)CC3)C	Not reported	
3	11,11-Dimethyl-spiro [2,9] dodeca-3,7-dien	Spiro compound	CC1(CC12CCC=CCCC=CC2)C	Not reported	
4	1H-3a,7-Methanoazulene, octahydro-1,9,9-trimethyl-4-methylene-, (1.α.,3aα.,7 α.,8aβ)	Sesquiterpene	CC1CCC23C1CC©(C)C)CCC3=C	Not reported	
5	1H-Cyclopropa[a]naphthalene. 1a,2,3,5,6,7,7a,7b-octahydro-1,1,7,7a-tetramethyl-, [1aR-(1aα,7α,7aα,7bα)]-	Sesquiterpene	CC1CCC=C2C1(©(C3(C)C)CC2)C	Antibacterial	[21]
6	1-Isopropyl-4,7-dimethyl-1,2,3,5,6,8a-hexahydronaphthalene	Sesquiterpene	CC1=CC2C(CCC(=©C1)C)C(C)C	Not reported	
7	2H-Pyran, tetrahydro-4-methyl-2-(2-methyl-1-propenyl)	Monoterpene	CC©OC(C1)C=C(C)C	Anti-inflammatory	[22]
8	3H-3a,7-Methanozulene, 2,4,5,6,7,8-hexahydro-1,4,9,9-tetramethyl-, [3aR-(3aα,4β,7α)]	Sesquiterpene	CC1CC©C3=C(CCC13C2(C)C)C	Not reported	
9	3-Methylmannoside	Carbohydrate	COC1C(C(OC(C1O)O)CO)O	Not reported	
10	3-O-Methyl-d-glucose	Carbohydrate	COC(C(C=O)O)C(C(CO)O)O	Anticancer and anti-inflammatory	[23]
11	5,7-Dodecadiyne-1,12-diol	Fatty acid	C(CCO)CC#CC#CCCCCO	Not reported	
12	1,5-Cyclodecadiene, 1,5-dimethyl-8-(1-methylethenyl)-, [S-(Z,E)]-	Terpenoid	CC1=CCCC(=CCC(CC1)C(=C)C)C	Not reported	
13	Caryophyllene	Sesquiterpene	CC1=C©(=C)C2CC(C2CC1)(C)C	Anticancer, antioxidant, and antimicrobial	[24]
14	Ergosta-5,22-dien-3-ol, acetate, (3β,22E)-	Sterol	CC(C)C(C)C=CC(C)C1CCC2C1(CCC3C2CC=C4C3(CCC(C4)O)C)C	Not reported	
15	Germacrene D	Sesquiterpene	CC1=CCCC(=C)C=CC(CC1)C(C)C	Mosquitocidal and repellent activity against aphids and ticks	[25,26,27]
16	Humulene	Sesquiterpene	CC1=CCC(C=CCC(=CCC1)C)(C)C	Anti-inflammatory, analgesic, and antineoplastic	[28]
17	Naphthalene, 1,2,4a,5,6,8a-hexahydro-4,7-dimethyl-1-(1-methylethyl)-	Sesquiterpene	CC1=CC2C(CC1)C(=CCC2C(C)C)C	Not reported	
18	α-Copaene	Sesquiterpene	CC1=CCC2C3C1C2(CCC3C(C)C)C	Antiproliferative and antioxidant	[21]
19	γ-Elemene	Sesquiterpene	CC(=C1CCC(C(C1)C(=C)C)(C)C=C)C	Insecticidal	[29]

**Table 3 plants-12-01495-t003:** Results of the binding interactions of the compound Ergosta-5,22-dien-3-ol, acetate, (3β,22E), with the five target proteins.

Phytocompound	Target Proteins	Binding Energy (∆G) (Kcal/mol)	Ligand Efficiency	Inhibition Constant ((Ki) (nM))
Ergosta-5,22-dien-3-ol, acetate, (3β,22E)	TNF-α	−8.21	−0.26	964.03
	Cox-2	−8.93	−0.28	282.86
	Superoxide dismutase	−7.16	−0.22	5.62
	α-Amylase	−9.27	−0.29	161.19
	Aldo reductase	−11.73	−0.37	2.53

**Table 4 plants-12-01495-t004:** Independent and dependent variables used in the Box–Behnken design for the optimisation of *Causonis trifolia* shoot extraction.

Symbols	Independent Variables	Unit	Coded Levels		
			−1 (Low)	0 (Medium)	+1 (High)
A	Methanol concentration	%	40	70	100
B	Temperature	°C	20	50	80
C	Extraction duration	Hours	1	12.5	24
	Dependent variable/response variable		Goal		
Y1	Total phenolic content	mg GAE/g dry tissue	Maximise		

**Table 5 plants-12-01495-t005:** Three-factorial RSM-based optimisation study using a Box–Behnken design.

Run	Solvent (%)	Temperature (°C)	Extraction Duration (Hours)	Total Phenolic Content (Mg GAE/g Dry Tissue)
1	70	20	24	100.24
2	70	20	1	108.45
3	100	50	24	103.88
4	70	50	12.5	154.35
5	70	50	12.5	151.25
6	70	50	12.5	155.84
7	40	50	24	108.68
8	70	80	24	105.34
9	70	80	1	100.23
10	40	50	1	105.22
11	100	80	12.5	100.55
12	70	50	12.5	150.44
13	70	50	12.5	151.33
14	40	80	12.5	116.45
15	40	20	12.5	111.22
16	100	20	12.5	107.45
17	100	50	1	102.22

## Data Availability

All data are contained within the article.

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
