# Peer review of "Phytochemical Profiling, Biological Activities, and In Silico Molecular Docking Studies of Causonis trifolia (L.) Mabb. & J.Wen Shoot"

_plants, 2023, doi:10.3390/plants12071495_

Round 1

Reviewer 1 Report

Full Title: Phytochemical profiling and pharmacological investigation of optimized phenolic extract from Causonis trifolia (L.) Mabb. & J.Wen shoot and molecular docking interaction of its bioactive compounds with target proteins

Manuscript Number: Plants 2022, 11, x. https://doi.org/10.3390/xxxxx

Article Type: Research article

General comments:

In the manuscript “Phytochemical profiling and pharmacological investigation of optimized phenolic extract from Causonis trifolia (L.) Mabb. & J.Wen shoot and molecular docking interaction of its bioactive compounds with target proteins”, authors attempted to investigate the anti-inflammatory, antioxidant, and antidiabetic activity of Causonis trifolia (Vitaceae) extracts based on it is ethnopharmacological information, chemical constituents and in silico molecular docking analysis. GC MS, FTIR and HPLC were used to investigate the chemical constituent of Causonis trifolia extracts.

Overall, the work is extensive and well presented. The manuscript contains a massive amount of experimental research work and results. However, the presentation of the material and methods and discussion parts should be improved. The font size and resolution for the figure and table should be unified. Figures and tables were occupied too much space in many places, they could be rearranged and reorganized.

Specific comments:

1.     Line 39, Line 394 and Line 749: “In vitro” should be unified with the italic font “In vitro”.

2.     Line 40, Line 406 and Line 753 : “In vivo” should be unified with the italic font “In vivo”.

3.     Line 65, Line 70, Line 565, Line 578, Line 584 and Line 705: delete the extra space.

4.     Line 197:  unify the font size in Table 2.

5.     Line 232-254: improve the presentation of the GC-MS methods to match well with the result. State clearly the source of the TMS in the detected compounds. If you have used derivatization method (e.g. silylation), it should be described precisely in the method part.

6.     Line 474: It might be better to place the heading of Table 3 (in page 10) closely to the Table 3 continents in page 11.

7.     Line 503, Line 517 and Line 700: Unify the font and improve the resolution in Figure 3, Figure 4 and Figure 10 Y and x axes respectively.

8.     Line 547 and Line 544: Fix the resolution of Figure 5 and Figure 6.

9.     Line 634: space between numbering in “3.8.1.α-. Amylase inhibitory assay” and why dot after alpha “1.α-. Amylase inhibitory”

10.  Line 674: unnecessary bold font for the name of the compounds.

11.  Line 678: space is missing in “kcal/mol),and diabetic”

12.  Line 694: Improve Table 7. In table 7, table capture is needed to identify the meaning of the bold text and abbreviations.

13.  Line 698: Improve Table 8; rewrite the name of the compounds to be in one or two row. Why the target proteins were in bold font, place a line after table headers.

14.  Line 704 and Line 713: Improve “Figure C” in Figure 11. A and Figure 11, B. Also, I recommend to rewrite the figures without dots after numbering as “Figure 11 A” and “Figure 11 B”. 

15.  Line 715 and Line 724: space is missing in “(b)Binding pose”

16.  Line 731 and Line 740: “Figure 13. A” and “Figure 13. B”, As I mentioned above, I recommend to rewrite them without dots after numbering as “Figure 13 A” and “Figure 13 B”. 

17.  Line 739: Improved “Figure C” in Figure 13. A and Figure 13, B.

18.  Line 794; correct the hyphen (-) sign in “(Figure 1-2)” to shorter en dash (–).

19.  Line 900: “PECTS” the font size should be corrected.

20.  Line 1055: References and acronyms should be checked and edited in a consistent manner. Considered the references format requested by the journal of antibiotics.

21.  Line 1087: remove the red font color in “anti inflammatory”

Author Response

From,                                                                                                                          

Prof. Chowdhury Habibur Rahaman

Department of Botany

Visva-Bharati University

Santiniketan, West Bengal, India

To,

The Reviewer,

Plants, MDPI

Dear sir/madam,

I would like to thank you for this opportunity to resubmit a revised copy of our manuscript entitled “Phytochemical profiling and pharmacological investigation of optimized phenolic extract from Causonis trifolia (L.) Mabb. & J.Wen shoot and molecular docking interaction of its bioactive compounds with target proteins” (Manuscript ID- plants-2170288). We, all the authors, would like to take this opportunity to express our gratitude for the positive feedback and insightful comments, which have greatly helped us to improve the quality of our manuscript.

In accordance to the comments, all major and minor changes in the text were marked up using the “Track Changes” function in the revised manuscript. Enclosed please find our revised manuscript and list of changes made point-by-point.

Hope our revised manuscript will satisfy you in all respect.

Regards,

Yours sincerely,

Prof. Chowdhury Habibur Rahaman

Reviewer 2 Report

Authors have performed phytochemical and pharmacological investigatin of PECTS extracts. There is no novelty in the research work, no compound was isolated from the extracts.

I have concerned with the following points

- Anti-oxidant and antiinflammatory activities of this plant is reported previously. Changing the part is not adding any novelty in the design of research work.

-There is no rational / logic to do FTIR for the extracts. I am not sure why the authors did it. FTIR spectra does not match with the compound's peak identified via LCMS and GCMS.

- The concentration of extracts used to test biological work dies not represent the doubling technique. Moreover, the lowest concentration used was 100microgram which is too high itself.

- Since structures are putative, in silico docking results do not carry any significance. It is also evident from the docking results.

- No validation of docking protocols were performed by docking x-ray bound ligands.

Author Response

(The authors gave the same response as above.)

Reviewer 3 Report

The manuscript is relatively large, even for an online publication. The authors should consider optimizing the text, particularly the introduction and sections concerning extraction optimization, to make it more compact. A large part of optimization data presented as figures and tables can also be moved to supplementary materials. 

The major problem with the experimental part of the manuscript is data based on GC-MS analyses. In the materials and methods, the authors did not mention the derivatization of the samples. Yet, the compound selected for in-silico docking study, identified as androstane-11,17-dione, 3-[(trimethylsilyl)oxy]-, 17-[O-(phenylmethyl)oxime], (3α,5α), does not occur naturally in plants. Absolutely all known compounds containing trimethyl silyl (TMS) groups are synthetics. If the authors did derivatize samples, then TMS groups would be present on all free hydroxyl groups, for example, in 3-methylmannoside. They are absent; therefore, the presence of this compound in Causonis trifolia extract is either a contamination or misidentification. In either case, such erroneous data puts suspicion on other identifications made by the authors. The structure-dependent docking data is also wrong if the structures are incorrect. Therefore, many of the results presented in the manuscript are invalidated.

The compound identified as benzenepropanoic acid, TBDMS derivative, has the same problem. It does not occur naturally; if the authors did not derivatize samples, it should not be present in the sample. 

As a long-time practicing natural products chemist, I struggle to find why the authors selected these compounds for molecular docking experiments.

Author Response

(The authors gave the same response as above.)

Reviewer 4 Report

The manuscript titled “Phytochemical profiling and pharmacological investigation of optimized phenolic extract from Causonis trifolia (L.) Mabb. & J.Wen shoot and molecular docking interaction of its bioactive compounds with target proteins” further deepen the knowledge on these extracts’ phytochemical profile and bioactivities.

In order to improve the manuscript, the authors should address some points, namely:

1.             Please revise the English language, grammar, and minor typos (e.g. italicize in vivo and in vitro; abbreviations should be described in the first appearance in the text).

2. Please revise sentences such as in lines 66/67. The authors refer to the study but not their conclusions or main results;

3. The introduction is extensive, please consider referring to methods details (e.g. line 93-97) in the methodologies section;

4. In section 2.3., the abbreviated format used to describe these methods is a valid option as these are common methods, but the authors could add the name of the method used, to help the reader understand the assay performed without consulting the bibliography. For example “total phenolic content was determined by method X, total flavonoid content by method Y”;

5. Line 212/213, please revise this sentence, is it relative to extracts or standards?

6. Line 241, units in filter size are missing;

7. Have the authors identified and quantified catechin or catechin hydrate? Please provide an HPLC chromatogram;

8. Figures 5 and 6 quality should be improved. In figure 6, all structures should be displayed using the same style concerning bond orientation, colour and structure style;

9. Please revise table 5 and 6 placement in text, part of the text could be used to separate the tables and improve the readabilit ;

10. Identify section 3.7.4.;

11. Regarding in silico analysis, for example the interaction with SOD is not sufficient to correlate to a cellular antioxidant activity, as this is only a single protein in a major cellular process:

12. Please provide the data for cytotoxicity assays;

13. For in vivo assays, did the authors perform the analysis of toxicity biomarkers or just physiological responses?

14. Lines 781/782, please revise. No phytochemicals were added, only the extraction was improved;

15. Line 783, should be specified that this sentence is valid for this plant;

16. More discussion should be added concerning the relevance of the in silico studies. The authors highlight a compound and proteins with the best binding, but thus this translates into health-promoting activities or toxicity? For example, when binding to SOD, if the compound inhibits the enzyme it should not be accounted as an antioxidant.

Overall, the manuscript presented here is very interesting with significant in vitro and in vivo assays. It can benefit from revising the various topics described and complementing the discussion.

Author Response

(The authors gave the same response as above.)

Reviewer 5 Report

The manuscript entitled "Phytochemical profiling and pharmacological investigation of optimized phenolic extract from Causonis trifolia (L.) Mabb. & J.Wen shoot and molecular docking interaction of its bioactive compounds with target proteins" has been reviewed and found that is a good designed with scientific soundness and interested for the readers, just needs to consider the following comments to improve the quality of the Manuscript.

1. The title is so long and suggested to shortened.

2. The intensive revision for the typewriting mistakes throughout the manuscript must be performed

3. In the section of Materials and Methods: the Model of the equipments, Manufacturer, City and Country must be presented

4. 2.7.1. DPPH radical scavenging activity mentioned that "DPPH radical scavenging activity was determined following the standard method 257 [26] with slight modifications." and "2.9.1.α-. Amylase inhibitory assay 329 The assay was carried out following the standard protocol with slight modifications, The authors asked to explain, which kind of modifications have been done?

5. In 3.4. High performance liquid chromatography (HPLC) analysis of PECTS, It is better to present the HPLC chromatogram

6. In the section of Results and discussion, the results must be deeply interpreted  and showing the hypothesis of the authors the scientific explanation as well as compare the results with previously published work.

7. The conclusion should focus on the main findings and novelty which supported with results, also i suggest to add the two sentences about limitations and Applications

8. Several references without DOI. DOI of publications must be added  

Author Response

(The authors gave the same response as above.)

Round 2

Reviewer 2 Report

Most of the querries are addressed by authors.

Author Response

From,                                                                                                                          

Prof. Chowdhury Habibur Rahaman

Department of Botany

Visva-Bharati University

Santiniketan, West Bengal, India

To,

The Reviewer,

Plants, MDPI

Dear sir/madam,

I would like to thank you for this opportunity to resubmit a revised copy of our manuscript entitled “Phytochemical Profiling, Biological Activities and In Silico Molecular Docking Studies of Causonis trifolia (L.) Mabb. & J.Wen Shoot” (Manuscript ID- plants-2170288). We, all the authors have tried to addressed all your queries and would like to take this opportunity to express our gratitude for the positive feedback and insightful comments, which have greatly helped us to improve the quality of our manuscript.

Hope our revised manuscript will satisfy you in all respect.

Regards,

Yours sincerely,

Prof. Chowdhury Habibur Rahaman

Most of the queries are addressed by authors.

- Your valuable suggestions and noteworthy inputs are highly appreciated and important for our manuscript. All the authors are very happy to know that we have successfully addressed your queries and satisfied you. Thank you very much for your sincere help.

Reviewer 3 Report

Unfortunately, the manuscript, despite the corrections, is still underdeveloped and is not suitable for publication in its present form.

The authors did not explain in the presented response whether the samples were derivatized, "problematic" compounds such as silyl derivatives or TBDMS derivative - generally were deleted from the manuscript without any explanation.

In addition, most of the presented figures are of very poor quality, especially Figure5. GC-MS chromatogram obtained from optimized phenolic extract of Causonis trifolia shoot, but not only. Text in some figures was illegible, please provide clear illustrations in higher resolution.

Author Response

From,                                                                                                                          

Prof. Chowdhury Habibur Rahaman

Department of Botany

Visva-Bharati University

Santiniketan, West Bengal, India

To,

The Reviewer,

Plants, MDPI

Dear sir/madam,

I would like to thank you for this opportunity to resubmit a revised copy of our manuscript entitled “Phytochemical Profiling, Biological Activities and In Silico Molecular Docking Studies of Causonis trifolia (L.) Mabb. & J.Wen Shoot” (Manuscript ID- plants-2170288). We, all the authors, would like to take this opportunity to express our gratitude for the positive feedback and insightful comments, which have greatly helped us to improve the quality of our manuscript.

In accordance to the comments, all major and minor changes in the text were marked up using the “Track Changes” function in the revised manuscript. Enclosed please find our revised manuscript and list of changes made point-by-point.

Hope our revised manuscript will satisfy you in all respect.

Regards,

Yours sincerely,

Prof. Chowdhury Habibur Rahaman

The authors did not explain in the presented response whether the samples were derivatized, "problematic" compounds such as silyl derivatives or TBDMS derivative - generally were deleted from the manuscript without any explanation.

- Sir during our study we have not performed any derivatization process.

According to the comments of the editor and reviewer 3, we have realised that the silyl derivatives or TBDMS derivatives detected in the GC-MS analysis are the impurities. These silyl derivative compounds are Androstane-11,17-dione, 3-[(trimethylsilyl)oxy]-, 17-[O-(phenylmethyl)oxime], (3α,5α); Benzenepropanoic acid, TBDMS derivative; Silane, triethyl(2-phenylethoxy); Silicic acid, diethyl bis(trimethylsilyl) ester. Hence, in our revised manuscript, we have eliminated the above-mentioned 4 compounds. Now, the final numbers of detected compounds in GC-MS analysis of the stem extract of the investigated plant are 19 (in our previous manuscript this number was 23).

In addition, most of the presented figures are of very poor quality, especially Figure5. GC-MS chromatogram obtained from optimized phenolic extract of Causonis trifolia shoot, but not only. Text in some figures was illegible, please provide clear illustrations in higher resolution.

- Thank you for mentioning this point. We have improved the resolution of the ‘Figure 1’, ‘Figure 2’ ‘Figure 5’ ‘Figure 7’ ‘Figure 8’ ‘Figure 9’ and ‘Figure 15’. Moreover, we have provided a separate .zip file, which contains all the figures in high resolution.

Round 3

Reviewer 3 Report

Accept in present form.